

# Willingness-to-pay for a probabilistic flood forecast: a risk-based decision-making game

Louise Arnal[1,2], Maria-Helena Ramos[3], Erin Coughlan[4], Hannah Louise Cloke[1,5], Elisabeth Stephens[1], Fredrik Wetterhall[2], Schalk-Jan van Andel[6], and Florian Pappenberger[2,7]

[1]Department of Geography and Environmental Science, University of Reading, Reading, UK
[2]ECMWF, European Centre for Medium-Range Weather Forecasts, Shinfield Park, Reading, UK
[3]IRSTEA, Catchment Hydrology Research Group, UR HBAN, Antony, France
[4]Red Cross/Red Crescent Centre, The Hague, the Netherlands
[5]Department of Meteorology, University of Reading, Reading, UK
[6]UNESCO-IHE Institute for Water Education, Delft, the Netherlands
[7]School of Geographical Sciences, University of Bristol, Bristol, UK

*Correspondence to:* Louise Arnal (l.l.s.arnal@pgr.reading.ac.uk)

**Abstract.** In order to communicate forecast uncertainty, there has been a gradual adoption of probabilistic hydro-meteorological forecasts. This uncertainty is twofold, as it constitutes both an added value and a challenge for the forecaster and the user of the forecasts. Many authors have demonstrated the added (economic) value of probabilistic forecasts over deterministic forecasts, for diverse activities of the water sector (e.g. flood protection, hydroelectric power management and navigation). However, the
richness of the information is also a source of challenges for operational uses, due partially to the difficulty to transform the probability of occurrence of an event into a binary decision. This paper presents the results of a risk-based decision-making experiment, set up as a game on the topic of flood protection mitigation, called "How much are you prepared to pay for a forecast?". The game was played at several workshops in 2015, which were attended by operational forecasters and academics working in the field of hydro-meteorology. The aim of this experiment is to contribute to understanding the role of
probabilistic forecasts in decision-making processes and their perceived value by decision-makers. Based on the participants' willingness-to-pay for a forecast, the results of the game show that the value (or the usefulness) of a forecast depends on several factors, including the way users perceive the quality of their forecasts and link it to the perception of their own performances as decision-makers. Balancing avoided costs and the cost (or the benefit) of having forecasts available for making decisions is not straightforward, even in a simplified game situation, and is a topic that deserves more attention from the hydrological
forecasting community.

## 1   Introduction

In a world where hydrological extreme events, such as droughts and floods, are likely to be increasing in intensity and frequency, vulnerabilities are also likely to increase (WMO, 2011; Wetherald and Manabe, 2002; Changnon et al., 2000). In this context, building resilience is a vital activity. This can be done by implementing early warning systems, of which hydrological forecasts
are key elements.



Hydrological forecasts suffer from inherent uncertainties, which can be from diverse sources, including: the model structure, the observation errors, the initial conditions (e.g. snow cover, soil moisture, reservoir storages, etc) and the meteorological forecasts - of precipitation and temperature - (Verkade and Werner, 2011; He et al., 2009). The latter variables are fundamental drivers of hydrological forecasts and are therefore major sources of uncertainty. In order to capture some of this uncertainty,

there has been a gradual adoption of probabilistic forecasting approaches, with the aim to provide forecasters and forecast users with additional information not contained in the deterministic forecasting approach. Whereas "a deterministic forecast specifies a point estimate of the predictand (the variate being forecasted)", "a probabilistic forecast specifies a probability distribution function of the predictand." (Krzysztofowicz, 2001). For operational forecasting, this is usually achieved by using different scenarios of meteorological forecasts following the ensemble prediction approach (Buizza, 2008; Cloke and Pappenberger,

10  2009).

Many authors have shown that probabilistic forecasts provide an added (economic) value compared to deterministic forecasts (Buizza, 2008; Verkade and Werner, 2011; Pappenberger et al., 2015). This is due to, amongst others, the quantification of uncertainty by probabilistic forecasting systems, their ability to better predict the probability of occurrence of an extreme event and the fact that they issue more consistent successive forecasts (Dale et al., 2012; Cloke and Pappenberger, 2009).

This probability of occurrence makes the probabilistic forecasts useful in the sense that they provide information applicable to different decision thresholds, essential since not all forecast users have the same risk tolerance (Michaels, 2015a; Buizza, 2008; Cloke and Pappenberger, 2009). Probabilistic forecasts therefore enable the quantification of the potential risk of impacts (New et al., 2007) and, as a result, they can lead to more optimal decisions for many hydrological operational applications, with the potential to realise benefits from better predictions (Verkade and Werner, 2011; Ramos et al., 2013). These applications

are, for example, flood protection (Stephens and Cloke, 2014; Verkade and Werner, 2011), hydroelectric power management (García-Morales and Dubus, 2007; Boucher et al., 2012) and navigation (Meissner and Klein, 2013). Moreover, the continuous increase in probabilistic forecast skill is very encouraging for the end-users of the probabilistic forecasts (Bauer et al., 2015; Magnusson and Källén, 2013; Simmons and Hollingsworth, 2002; Ferrell, 2009).

However, the communication of uncertainty through probabilistic forecasts and the use of uncertain forecasts in decision-

making are also challenges for their operational use (Cloke and Pappenberger, 2009; Ramos et al., 2010; Michaels, 2015a; Crochemore et al., 2015). One of the reasons why the transition from deterministic to probabilistic forecasts is not straight-forward is the difficulty in transforming a probabilistic value into a binary decision (Dale et al., 2012; Demeritt et al., 2007; Pappenberger et al., 2015). Moreover, decision-makers do not always understand probabilistic forecasts the way forecasters intend them to (Handmer and Proudley, 2007). This is why it is essential to bridge the gap between forecast production and hazard

mitigation, and to foster communication between the forecasters and the end-users of the forecasts (Cloke and Pappenberger, 2009; Michaels, 2015a).

As Michaels (2015a, b) notes, "the extent to which forecasts shape decision making under uncertainty is the true measure of the worth of a forecast". The potential added value of the forecast can furthermore only be entirely realised with full buy-in from the decision-makers. However, how much are users aware of this added value? How much are they ready to pay for a

forecast? These are questions that motivated the work presented in this paper. In order to understand how users perceive the





value of probabilistic forecasts in decision-making, we designed a risk-based decision-making game - called "How much are your prepared to pay for a forecast?" - focusing on the use of forecasts for flood protection. The game was played during the European Geophysical Union (EGU) General Assembly meeting 2015 (Vienna, Austria), at the Global Flood Partnership (GFP) workshop 2015 (Boulder, Colorado), as well as at Bristol University (BU) in 2015. Games are increasingly promoted and used

to convey information of scientific relevance. They foster learning, dialogue and action through real-world decisions, which allow to study the complexities hidden behind real-world decision-making in an entertaining and interactive setup (de Suarez et al., 2012).

    This paper presents the game elaborated and the results obtained from its different applications. We propose to analyse the participants' perceived forecast value by investigating the way participants use the forecasts in their decisions and their

willingness-to-pay (WTP) for a probabilistic forecast. The WTP is the amount an individual is inclined to disburse to acquire a good or a service, or to avoid something undesirable (Breidert et al., 2006; Leviäkangas, 2009). It is a widely and very commonly adopted method to make perceived value assessments and its use has been demonstrated in a meteorological context (Leviäkangas, 2009; Anaman et al., 1998; Rollins and Shaykewich, 2003; Breidert et al., 2006). Breidert et al. (2006) present a complete overview of the methods available, organised according to data collection types. According to their classification,

there exists two main WTP measuring approaches: the "revealed preference" and the "stated preference" one. The former describes price-responses methods (such as market data analysis, laboratory-experiments and auctions, amongst others) while the latter refers to surveys in general. This experiment combines both "revealed preference" and "stated preference" methods. The design of the game is described in Section 2 and justified in terms of the purpose and contribution of the different components of the game to its main aim. The results and the discussion promoted by the latter are subsequently presented in Sections 3 and

4, respectively.

## 2   Set up of the decision-making experiment

### 2.1   Design of the experiment

This game was inspired by the table game "Paying for Predictions", designed by the Red Cross/Red Crescent Climate Centre[1]. Its focus is however different. Here, our aim is to investigate the use of forecasts for flood protection and mitigation. Also, we

strongly adapted the game to be played during conferences and with large audiences.

    The setup of the game was the following: participants were told they were competing for the position of head of the flood protection team of a company. Their goal was to protect inhabitants of a fictitious town bordering a fictitious river against flood events, while spending as little money as possible during the game. The participant with the highest amount of money at the end of the game was chosen as head of the flood protection team. Each participant was randomly assigned a river (river

yellow, river blue or river green) for the entire duration of the game. Each river had distinct initial river levels and rate of flood

---

[1]http://www.climatecentre.org/resources-games/paying-for-predictions



occurrences (see Table 1). Participants worked independently and had a worksheet to take notes (see Appendix). An initial purse of 20,000 tokens was given to each player to be used throughout the game.

Based on this storyline, the participants were presented the following sequence of events (illustrated by Fig. 1, left): after being given their river's initial level (ranging between 10 to 60 included), each participant was asked to make use of a prob-
abilistic forecast (see Fig. 1, right) of their river level increment after rainfall (ranging between 10 to 80 included) to decide if they wanted to pay for flood protection or not. The cost of flood protection was 2,000 tokens. They were informed, prior to the start of the game, that a flood occurred if the sum of the initial river level and the river level increment after rainfall (i.e. the actual river level after rainfall) reached a given threshold of 90. The probabilistic forecasts were visualised using boxplot distributions. They had a spread of about 10 to 20, and indicated the $5^{th}$ and $95^{th}$ percentiles as well as the median (i.e. $50^{th}$
percentile) and the lower and upper quartiles (i.e. $25^{th}$ and $75^{th}$ percentiles respectively) of the predicted river level increment after rainfall. Forecasts, as shown in Fig. 1 (right), were given to participants case by case (i.e. when playing the first case, they could only see the boxplot distribution of forecast river increment for case 1). Once the participants had made their decisions using both pieces of information (i.e. river level before rainfall and forecast of river level increment), they were given the observed (actual) river level increment after rainfall for their rivers. If a flood occurred and the participant had not bought flood
protection, a penalty of 4,000 tokens had to be paid.

The monetary values (initial purse, price of flood protection and penalty for not having protected the city when a flood occurred) were deliberately chosen. The price of a protection was set to 2,000 tokens such that if a participant decided to buy flood protection every time during the game (i.e. two rounds of five cases each, thus ten times) they would have no tokens left in their purse at the end of the game. This was done in order to discourage such a behaviour. The damage price, price paid when
no protection was bought against a flood that actually happened, was set to twice the flood protection cost as this was estimated to be a realistic relation between the two prices based on Pappenberger et al. (2015). The latter states that the avoided damages due to early flood warning amounts to a total of about forty percent. Here, for simplicity, we used a percentage of fifty percent.

Once the context explained, the participants were then told that they would play one round of five independent cases, which would each be played exactly according to the sequence of events afore presented, and for which they would have to record
their decisions on the worksheet they were provided (see Appendix). The game had in total two rounds of five cases each. This specific number of cases and rounds was chosen because of the time-constraint to play the game during conferences (the game should last around 20 to 30 minutes only). The total number of flood events was purposely kept the same for each river, in order to give equal chances to all participants to win the game. Table 1 presents the total number of flood events for each round and each river. The number of flood events was different for every river for each round as river level values were randomly
generated for the purpose of the game.

During the first round of the game, the participants had forecasts of river level increments to help their decisions. These forecasts were however not available for all participants in the second round, but were sold between the two rounds through an auction. The purpose and setup of each round and the auction are explained in the following paragraphs.





### 2.1.1 Round 1

The objective of the first round was to familiarise the participants with the probabilistic forecasts they were given to help them in their decisions, and to create a diversity amongst the decision-makers in terms of:

- their river behaviour: which is why different rivers, each with different flood frequencies and different initial levels, were
assigned to the participants;

- the money they would spend during this round and have in hand for the ensuing auction (before round 2);

- the quality of their forecasts in the first round: to this end, different forecast sets were distributed to the players for round 1.

This diversity was triggered in round 1 in order to analyse whether or not the WTP for a second forecast set, measured in
the auction performed before round 2, was dependent on any of the factors inherent to the first round (i.e. river-specific flood frequency, money left in purse, or quality of the forecasts).

Before the start of the first round, each participant was given a forecast set, containing probabilistic forecasts of their river level increment after rainfall for the five cases of round 1. Participants were however not aware that three different forecast sets were produced for each of the rivers. One set had only forecasts with a positive bias (forecast sets 1), one set had only
unbiased forecasts (forecast sets 2) and one other set only forecasts with a negative bias (forecast sets 3). There were therefore nine different sets of forecasts, which were distributed randomly amongst the audience prior to the start of the game. The unbiased forecasts (forecast sets 2) had the observations fall between the lower and the upper quartiles of their distributions, while the biased forecasts (forecast sets 1 and 3) had the observations fall outside of the lower and the upper quartiles of their distributions.

The quality of each forecast set can be represented in terms of the number of correct forecast flood events (given a forecast percentile threshold), with respect to the number of observed flood events. For each forecast set type and each river, the number of forecast flood events during the first round was calculated by adding the median of the forecast river level increment to the initial river level for each case. A forecast is referred to as a false alarm if this sum forecasts a flood but the flood is subsequently not observed. It is referred to as a hit if the sum forecasts the flood (i.e. it exceeds the flood threshold) and
the flood is subsequently observed. A miss is an observed flood that was not forecast. The numbers of hits, misses and false alarms are usually gathered in a contingency table as a matrix (e.g. Table 2): hits are placed on top, left; misses on bottom, left, and false alarms on top, right. The place on bottom, right is usually not considered in the evaluation of forecasts as it represents situations with low interest to a forecaster (i.e. when floods are nor forecast, nor observed). Table 2 displays the nine contingency tables we obtain considering each forecast set type and each river. Each participant would find themselves in
one of the contingency tables represented. We can see the higher number of total misses (false alarms) considering all rivers together in forecast set type 3 (type 1) and the absence of these in the unbiased forecast set type 2.



After all the five cases of round 1 were played, participants were asked to rate their performance as a decision-maker and the quality of their forecast set for round 1, on a scale from 'very bad' to 'very good' (the option 'I don't know' was also available) (see Appendix).

### 2.1.2 Auction

The auction was carried out after round 1 in order to measure the participants' WTP for a second forecast set, and to evaluate its dependencies on any of the elements of the game in round 1. The auction was implemented as follows.

At the end of the first round, participants were told that a second round of five independent cases would be played. They were asked to transfer the remaining tokens from round 1 to the second round. They were then told that the forecasting centre distributing the probabilistic forecasts now wanted the decision-makers to pay for the forecast sets if they wanted to have access
to them for the second round. Furthermore, they were informed that only thirty percent of them could get a second forecast set for this round. This percentage was chosen in order to restrict the amount of participants that could buy a forecast set (and create a competitive auction), while keeping a high enough number of participants playing with a forecast set in round 2 for the analysis of the results.

Participants were then asked to make a sealed bid, writing down on their worksheets the amount of tokens they were willing
to disburse from their final purse of round 1 to obtain a set of probabilistic forecasts for all the five cases of round 2. After the bids were made, a forecast set was distributed to the participants within the highest thirty percent of the bids. This was done through an auction. It was carried out by asking the participants if any of them wrote down a bid superior or equal to 10,000 tokens. If any participants did, they raised their hands, after which a forecast set - for the same river as the river assigned to them at the beginning of the game - was given to them. The auction continued by lowering the amount of tokens stated to the
participants until all forecast sets for round 2 were distributed. Each participant having bought a forecast set for round 2 then was asked to disburse the amount of tokens they paid for this forecast set from their remaining purse from round 1.

We note that participants were not told that the forecasts for the second round were all unbiased forecasts. Once again, the quality of the forecasts was kept secret in order for the participants to assign a value to the second forecast set that would strictly be related to the conditions under which they played the first round.

### 25  2.1.3 Round 2

The second round was played in order to measure the added value of an unbiased forecast set, compared to no forecast set at all, to the decisions of the participants on protecting or not against floods. Moreover, as the winner of the game was determined by the amount of tokens left in their purse at the end of the game, this round would give a chance to participants who bought a second forecast set to make up for the money spent with the auction, during round 2.
The second round developed similarly to the first round, with five independent cases of decision-making, with the exception that only participants who bought a second forecast set could use it to make their decisions. Participants who did not buy a second forecast set did not have any forecasts on which to base their decisions.





Subsequent to the five cases, the participants were asked to once again answer a set of questions (see Appendix). They were asked to rate their performance as a decision-maker in the second round, on a scale from 'very bad' to 'very good' (the option 'I don't know' was also available). Participants without a second forecast set were invited to provide a justification for not purchasing a set of forecasts for this round. Participants who had bought a second forecast set were also asked to rate the

quality of their forecast set for round 2 (on a scale from 'very bad' to 'very good', the option 'I don't know' was also available) and if those were worth the price they had paid for them. If not, they were asked to provide a new price that they would have rather paid.

The winner was finally determined by finding the player with the largest amount of tokens in their purse at the end of the game.

**2.2 Objectives and evaluation strategy**

The main aim of this paper is to investigate the participants' WTP for a probabilistic forecast set in the context of flood protection, following the game-experiment designed as presented in the previous paragraphs. It unfolds into two objectives that were pursued in the analysis of the results:

1. to analyse how participants used the information they were provided (probabilistic forecast sets) in this risk-based
decision-making context, and

2. to characterise the participants' WTP for a probabilistic forecast set for flood protection.

We assess these objectives through six questions, which are presented below, together with the evaluation strategy implemented.

**2.2.1 Did the participants use their forecasts and, in this case, follow the $50^{th}$ percentile of their forecast during the**
**decision-making process?**

This first question was investigated using the results of the first round. We first wanted to know if the players were actually using their forecasts to make their decisions. Moreover, we searched for clues indicating that the participants were following the $50^{th}$ percentile (i.e. the median) of the probabilistic forecasts. This was done in order to see if the $50^{th}$ percentile was considered by the players as the optimal value to use for the decision-making process under this specific flood risk experiment.

Additionally, this question relates, as mentioned earlier, to an intrinsic characteristic of the use of probabilistic forecasts for decision-making, which is the difficulty to transform the probabilistic values into a binary decision (Dale et al., 2012; Demeritt et al., 2007; Pappenberger et al., 2015). The way in which probabilistic flood forecasts are used depends on attitudes of decision-makers towards risk, uncertainty and the error in the information provided to them (Demeritt et al., 2007; Ramos et al., 2013), and decisions can vary from a participant to the next provided the same information (Crochemore et al., 2015).

Question one was explored by looking at the worksheets collected in order to infer from the decisions taken by the participants whether or not they most probably used the median of their forecasts to consider if the river level would be above, at or under the flood threshold. In cases where the decisions did not coincide with what the median forecast indicated, other factors



that could also influence the decisions were considered, such as: a) the flood frequency of each river and their initial river levels, b) the forecast set type each participant had (i.e. biased - positively or negatively - or unbiased) and c) the familiarity of the participants with probabilistic forecasts and decision-making (given their occupation and years of experience).

### 2.2.2 Was there a correspondence between the way participants perceived the quality of their forecasts in round 1 and their 'true' quality?

A well-known effect, called the "crying wolf", was studied for weather-related decision-making by LeClerc and Joslyn (2015). It describes the reluctance of users to comply with future alarms when confronted in the past with false alarms. This leads to the second question which was explored in this paper: was there a correspondence between the way participants perceived the quality of their forecasts in round 1 and their 'true' quality? Our aim here is to investigate whether the participants were more

sensitive to false alarms or misses. The participants' answers to the question on their forecast set quality for the first round (see Appendix) were analysed against their 'true' quality. The latter was measured in terms of forecast bias, calculated from the hits, false alarms and misses presented in Table 2. A bias value was computed for each forecast set type of each river (i.e. each contingency table; there were therefore nine different bias values in total) with the following equation:

$$Bias = \frac{hits \ + \ false \ alarms}{hits \ + \ misses} \tag{1}$$

A bias value equal to one is a perfect value, and a value less than (superior to) one indicates under (over) prediction.

### 2.2.3 Did the participants' perceptions of their own performance coincide with their 'true' performance?

We also looked at the perception the participants had of their own performance. The answers to the question "How was your performance as a decision-maker" (see Appendix) was assessed against the participants' 'true' performances (in rounds 1 and 2), which were calculated in terms of the money participants spent following the consequences of their decisions. The following

general formula (*n* being the round number) was used:

$$Performance = \frac{Money \ spent \ round \ n}{Optimal} \tag{2}$$

The performance is expressed relatively to the optimal performance, which is the minimum amount a participant could have spent, given the river they were assigned, defined as:

$$Optimal = Protection \ cost \times Number \ of \ floods \ in \ round \ n \tag{3}$$

A performance value of one indicates an optimal performance. Performance values greater than one indicate that participants spent more money than the minimum amount necessary to protect the city from the observed floods. The greater the value, the higher the amount of money unnecessarily spent.





### 2.2.4 What was the participants' willingness-to-pay for a probabilistic forecast set?

The auction was incorporated to the game in order to explore the WTP of participants for a probabilistic forecast set, considering the risk-based decision-making problem proposed by the game. To characterise this WTP, the bids were analysed and their relationship with several other aspects of the game were explored to explain the differences (if any) in the bids. These aspects were:

- the way participants used the forecasts. Here we try to learn about the effectiveness of the information on the user, which is an attribute of the value of information (Leviäkangas, 2009). It is considered that a participant is not expected to be willing to disburse any money for an information they are not using. The answers to question one (i.e. "Did the participants use their forecasts and, in this case, follow the $50^{th}$ percentile of their forecast during the decision-making process?") are used here.

- the money available to participants after round 1 to make their bids. As participants were informed at the beginning of the game that the winner would be the player with the highest amount of tokens in purse at the end, the tokens they had in hand for the auction (after round 1) may have restricted them in their bids. The bids are thus also explored considering the amount of tokens in hand at the time of the auction.

- the forecast set type. The bias of the forecasts during round 1 could also have been a potential determinant of participants's WTP for a forecast set in round 2.

- the river flood frequency. This was different for all the rivers in the first round and could be an element of the relevance of the information, another attribute of the value of information (Leviäkangas, 2009). Indeed, one could ask: "If my river never floods, why should I pay for forecasts?".

- the years of experience and occupation. This might influence the familiarity participants may have with the use of probabilistic forecasts for decision-making.

### 2.2.5 Did participants with a forecast set perform better than those without?

Round 2 was led by a central question: did participants with a forecast set perform better than those without? It was investigated by looking at the performance of participants in round 2, calculated using Eq. (2). While we expect players with more (unbiased) information to make better decisions, other factors could have influenced the trust participants had in the information during round 2, such as, for instance, the quality of the forecasts experienced by participants in round 1 or the flood events observed in the river in round 2, compared to the experience participants had previously had in round 1.

### 2.2.6 What were the winning and the losing strategies (if any)?

Finally, from the final results of the game, a question arose: what were the winning and the losing strategies (if any)? This question was explored by looking at the characteristics and decisions of the participants during the game, in order to distinguish





common attributes between these opposite strategies. Furthermore, an 'avoided cost' was calculated for each river based on the difference between the tokens spent by participants without a second forecast set and the tokens spent by participants with a second forecast set, during round 2. It represents the average amount of tokens participants without a second forecast set lost by protecting when a flood did not occur or by not protecting when a flood did occur, compared to participants with a second

forecast set. This 'avoided cost' was measured and compared to the average bid of participants for each river. An average 'new bid' was also calculated by replacing the bids of participants who had said that their forecast set in the second round was not worth the price they had paid initially, by the new bids they would have rather paid (see Appendix). This average 'new bid' was compared to the 'avoided cost' and the actual average bid obtained from the auction.

## 3   Results

The results are based on the analysis of 129 worksheets, from the 145 worksheets collected. The remaining 16 worksheets were either incomplete or incorrectly completed and were thus not used. Table 3 shows the distribution of the 129 worksheets, among the three forecast set types and the three rivers.

The game was played at the different events mentioned in the introduction. The participants present at those events displayed a diversity in terms of their occupation and years of experience. This was surveyed at the beginning of the game and is presented

in Fig. 2, for all the participants as well as for each river and forecast set type separately. Participants were mainly academics (postdoctoral researchers, PhDs, research scientists, lecturers, professors and students), followed by professionals (forecasters, operational hydrologists, scientists, engineers and consultants). The majority had less than five years of experience.

### 3.1   Participants were using the forecasts, but consistent patterns of use are difficult to detect

Figure 3 presents, on the one hand, the final purses of all the participants at the end of round 1, according to their river and

forecast set type (rows and columns respectively), and, on the other hand, the final purses that participants would have had if they had made their decisions according to the median of their forecasts. Participants in charge of the yellow river (upper row) ended the first round with, on average, more tokens than the others. Participants playing with the blue river (lower row) are those who ended round 1 with less money in purse, on average. This is due to the higher number of flood events for the blue river in round 1 (see Table 1). There are also differences in terms of final purses for the participants assigned the same river but

given a different forecast set type. Overall, participants who had unbiased forecasts (forecast set 2, middle column) ended the first round with on average more money than the other players. These results are an indication that the participants were using their forecasts to make their decisions.

In order to see if they were using the median values of the forecasts, a forecasted final purse was computed considering the case where the participants followed the median of their forecasts for all the cases of the first round (red vertical lines shown

on Fig. 3). If they had followed the median values of the forecasts during the entire first round, their final purses would have been equal to this value. Although this is almost the case for participants with forecast set type 2 (for all rivers), for participants



with the yellow river and forecast set type 1 and the green river and forecast set type 3, it is not an overall general observed behaviour.

Could some participants have discovered the bias in their forecasts and adjusted them for their decisions? Although it is hard to answer this question from the worksheets only, some of the decisions taken seem to support this idea. Figure 4 presents in more detail the results for the blue river in the first round. The forecast final levels are shown as boxplots for each forecast set type and for each of the five cases of round 1. These are the levels the river would reach if the initial level is added to the percentiles of the forecasts for each case. The bars at the bottom of the figure show the percentages of participants whose decisions differed from what the median of their forecast final level indicated [i.e. participants who bought (or did not buy) protection while no flood (or a flood) was predicted by the median of their forecast].

When comparing cases 1 and 4, for which the initial river levels and the observed and forecast final river levels were the same, we would not expect any changes in the way participants were using their forecasts. This is however not true. Figure 4 shows that the percentages of participants not following their forecast median differs between the two cases. For instance, about eighty percent of the participants with negatively biased forecast sets (forecast set type 3; under-predicting the increment of the river level) did not follow the median forecast in case 1, and did not protect against the predicted flood by their median forecast, while this percentage drops to about twenty percent in case 4. The fact that they were not consistently acting the same way may be an indication that they found out the bias in the forecasts and tried to compensate for it throughout round 1. We can also see that, in general, the lowest percentages of participants not following the median forecast are for forecast set type 2, which were unbiased. This is especially observed in the cases where the forecast final levels given by the median forecast are well above or below the flood threshold (cases 1, 2, 4 and 5). The fact that from case 1 to case 4, for forecast set type 2, we moved from about ten percent of participants not following the median forecast to zero percent, may also indicate that they built confidence in their forecasts (at least in the median value) along round 1, by perceiving that the median forecast could be a good indication of possible flooding or not in their river.

Figure 4 also shows that some participants with unbiased forecasts (forecast set type 2) did not always follow the median of their forecasts (for instance, cases 1, 3 and 5). Additional factors may therefore have influenced the way participants used their forecasts. A few worksheets indicated that these factors could be the distance of the median forecast or the distance of the initial river level to the flood threshold. In a few cases where the median forecast clearly indicated a flood, while the initial river level was low, some players did not purchase any flood protection. This can be observed on Fig. 4 for case 1, for example, for participants with forecast set type 1 or 2. The inverse situation (i.e. the initial river level was high, but the river level forecast by the median was low, below the flood threshold) was also observed and is illustrated on Fig. 4 for case 2 and forecast set type 3. Hence, in some cases, the initial river level seemed to also play a role in the decisions taken.

Other possible influencing factors, such as occupation and years of experience, were also investigated (not shown). No strong indication that these factors could have played a role in the participants' decision-making were however found.



### 3.2 Participants were overall less tolerant to misses than to false alarms in round 1

Figure 5 displays the cumulative percentages of participants having answered that the quality of their forecast set in round 1 (see Appendix) was 'very bad' (rating 1) to 'very good' (rating 5), as a function of the 'true' quality of the corresponding forecast, measured by the forecast bias (Eq. (1)). While participants with forecast sets for which the bias equalled one mostly rated their

forecasts 'quite good' or 'very good' (rating 4 and 5, respectively), the percentage of negative perceptions of the quality of the forecasts increases with increasing or decreasing forecast bias. It is interesting to note that participants with forecasts biased towards over-prediction, never rated their forecasts as 'very bad' (rating 1). Forecasts exhibiting under-prediction, seem to be less appreciated by the participants in terms of their overall quality. This can be an indication that participants were less tolerant to misses, while they accepted better forecasts leading to false alarms (over-predictions). This is contrary to the "crying wolf"

effect, and could be explained by the particular game setup, for which the damage cost (4,000 tokens) was twice the protection cost (2,000 tokens).

### 3.3 Participants had a good perception of their good (or bad) performance during the game and related it to the quality of their forecasts

Figure 6(a) illustrates the answers to the question "How was your performance as a decision-maker in round 1?" as a function

of the participants' 'true' performance (calculated from Eq. (2); i.e. the ratio to an optimal performance). The figure shows the number of participants for each perceived-actual performance combination, all rivers and forecast set types combined. The perceived decision-maker performance is presented on a scale from 1 ('very bad') to 5 ('very good'). An overall positive relationship between the participants' perceived performance and their 'true' performance is observed: the best performances are indeed associated with a very good perception of the performance by the decision-makers, and vice-versa. The same

analysis carried out for the answers concerning round 2 (not displayed) showed similar results: the rates participants gave to their performance were similarly close to their 'true' performance.

Figure 6(b) looks at the relationship between the perceived decision-maker performance and the rating the decision-makers gave to their forecast set quality in round 1. A positive relationship can also be seen: the majority rated their performance and the quality of their forecast set as 'quite good' (4) and 'very good' (5), while those who rated their performance 'very

bad' (1), also considered their forecast set 'very bad'. Therefore, the rating participants gave to their performance was closely connected to the rating they gave to their forecast set quality. This also contributes to the evidence that participants were using their probabilistic forecast sets to make their decisions. It is furthermore an indication that participants linked good forecast quality to good performance in their decision-making, and vice-versa.

### 3.4 Several factors may influence the WTP for a forecast, including forecast quality and economic situation

Given the evidence that most participants were using their forecasts to make their decisions in round 1 (see Section 3.1), we now investigate their willingness-to-pay (WTP) for a new forecast set to be used in round 2.





Figure 7 shows the bids participants wrote on their worksheets prior to the auction, for a second forecast set, as a function of the amount of tokens they had in their purses at the end of round 1. All bids are plotted and those from participants who succeeded in buying a second forecast set are displayed as red triangles on the figure. On average, participants were willing to pay 4,566 tokens, which corresponds to thirty-two percent of the average amount of tokens left in their purses. The minimum

bid was zero tokens (i.e. no interest in buying forecasts for round 2), which was made by ten percent of the players. Half of these players were participants who were assigned the blue river (the river for which players ended the first round with on average the lowest amount of tokens in purse). The only three participants who never bought flood protection in the first round (i.e. who could be seen as 'risk-seeking' players) made bids of zero, 3,000 and 4,000 tokens. The highest bid made was 14,000 tokens, corresponding to a hundred percent of the tokens left in that participant's purse. However, this participant did not raise

their hand during the auction to purchase a second forecast set. Nine participants (less than ten percent of the total number of players) made a bid of 10,000 tokens or above, corresponding to, on average, seventy-seven percent of the tokens they had left in their purses. The total cost of protecting all the time for round 2 being 10,000 tokens, as indicated on Fig. 7 by the dashed black line, bidding 10,000 tokens or more for a second forecast set was clearly pointless. Half of these participants were players to which the yellow river was assigned (the river for which participants finished the first round with on average the highest

amount of tokens in purse) and eight out of these nine participants had a forecast set with a bias during the first round. These nine participants, who paid 10,000 tokens or more for the second forecast set, were removed from the subsequent analyses of the auction results, as their bids suggest that they have not understood the stakes of the game.

From Fig. 7, there is a clear positive relationship between the maximum bids within each value of tokens left in purse and the tokens left in purse, as the participants did not disburse more tokens than they had left in their purse during the auction.

When we look at the evolution of the median of the bids with the amount of tokens in purse, in general, the more tokens one had left in purse, the higher their WTP for a forecast set. Nonetheless, the WTP seems to have a limit. It can be seen that from a certain amount of tokens left in purse on, the median value of the bids remains almost constant (in our game case, at about a bid of 6,000 tokens for participants with 12,000 tokens or more in their purse). The amount of tokens that the participants had in hand therefore only influenced to a certain extent their WTP for a second probabilistic forecast set.

We investigated if the way participants perceived the quality of their forecast set in the first round was a plausible determinant of their WTP for another forecast set to be used in round 2. Figure 8 shows the % bids, bids expressed as a percentage of the tokens participants had left in purse at the time of the auction, as a function of the rating participants gave to their forecast set quality in round 1 (from 1, 'very bad', to 5 'very good'; see Appendix). Firstly, it is interesting to observe that three participants judged their first forecast set to have been of 'very bad' quality (rating 1) but were nonetheless willing to disburse on average

fifty percent of the tokens they had left in purse. Those bids were however quite low, 4,000 tokens on average. Moreover, players who rated their first forecast set from 'quite good' to 'very good' (rating 4 and 5, respectively) were on average willing to disburse a larger percentage of their tokens than candidates who rated their previous forecast set from 'quite bad' to 'neither good nor bad' (rating 2 and 3, respectively). Therefore, the way participants rated the quality of their first forecast set was to a certain degree influential on their WTP for a second forecast set.



During the auction following the closed bids, forty-four forecast sets were distributed to the participants who made the highest bids, in order to be used in round 2. Table 4 shows that participants who purchased these second forecast sets were quite well distributed among the different forecast set types of round 1, with a slightly higher frequency of buyers among participants who had played round 1 with unbiased forecasts (forecast set type 2). Buyers also pertained more often to the group assigned river green (forty-eight percent), followed by river yellow (thirty-two percent) and blue (twenty-one percent). The buyers of the second forecast sets are displayed as red triangles on Fig. 7. We note that these red triangles are not necessarily the highest bid values on the figure, since we plot results from several applications of the game (in one unique application, they would coincide with the highest bids, unless a participant had a high bid but had not raised their hand during the auction to buy a second forecast set). Differences in the highest bids among the applications of the game can be an indication that the size (or type) of the audience might have had an impact on the bids (or on the WTP for a probabilistic forecast). Our samples were however not large enough to analyse this aspect.

Participants who did not purchase a second probabilistic forecast set (eighty-five players in total) stated their reason for doing so. The majority of them (sixty-six percent, or fifty-six players) said that the price was too high (which means, in other words, that the bids made by the other participants were too high, preventing them from purchasing a second forecast set during the auction). Ten participants (twelve percent) argued that the model did not seem reliable. Most of these participants were among those who had indeed received a forecast set with a bias in the first round. The rest of the candidates who did not purchase a second forecast set (twenty-two percent, or nineteen players) wrote down on their worksheet the following reasons:

– *Low flood frequency in the first round* - a participant assigned the yellow river wrote: "Climatology seemed probability of flood = 0.2".

– *Assessment of the value of the forecasts difficult* - a participant wrote: "No information for the initial bidding line" and another wrote: "Wrong estimation of the costs versus benefits".

– *Preference for taking risks* - "Gambling" was a reason given by a player.

– *Enough money left in purse to protect all the time during round 2* - which can be an indication of risk-averse behaviour coupled with economic wealth and no worries of false alarms.

– *Not enough money left in purse to bid successfully* - a participant wrote: "The purse is empty due to a lot of floods".

### 3.5 Decisions are better when they are made with the help of unbiased forecasts, comparatively to having no forecasts at all

The analysis of the results of round 2 allowed us to compare the performance of participants with and without a forecast set. Overall, participants without a second forecast set had an average 'true' performance value of 3.1, computed as shown in Eq. (2) and over the five cases of round 2. The best performance was equal to the optimal performance ('true' performance value equal to 1) and the worst performance reached a value of 6. Comparatively, participants with a second forecast set had an average 'true' performance of 1.2, thus much closer to the optimal performance than the average performance of participants without





a second forecast set. The best performance in this group also equalled the optimal performance, while the worst performance value was 2.5, much lower (i.e. thus much closer to the optimal value) than the worst performance value of participants making their decisions without any forecasts. These numbers clearly indicate that the possession of a forecast set in the second round led to higher performances and to a lower spread in performances within the group of players with a second probabilistic

forecast set (comparatively to players without forecasts in round 2).

All the participants without a second forecast set who were assigned the yellow river missed the two first floods in the second round. Part of these participants protected for all or some of the subsequent cases, while the other part never bought any protection. It could have been due to the low flood frequency of their river in the first round (see Table 1). This behaviour was not observed for the green river participants without a second forecast set, for which a very diverse sequence of decisions

was seen in the second round. As for the blue river participants without any second forecast set, most of them missed the first flood event that occurred in round 2 and, subsequently, protected for a few cases where no flood actually occurred. These decision patterns were not observed for participants with a second forecast set within each river, who took more consistently right decisions.

The large majority of participants with a second forecast set in round two (forty-one out of forty-four) rated their forecasts

either 'quite good' or 'very good', which was expected since all the forecasts were unbiased in round two.

## 3.6   Overall winning strategies would combine good performance and knowledge of the monetary value of a forecast set, with regards to costs of bad decisions

The average final purse at the end of round 2 was 3,149 tokens, remaining from the 20,000 tokens initially given to each participant. The minimum final purses observed were zero token or less. Twenty-five participants, out of the total 129 players,

finished the game with such amounts of tokens. Out of these twenty-five participants, twenty-two had received a biased forecast set in the first round. From the analysis of the game worksheets, we could detect at least three losing strategies followed by these twenty-five participants who finished with zero token or less in purse:

1. Eighteen participants, most of them blue river players, had an 'acceptable to bad' performance in round 1, did not purchase a second forecast set, and performed badly in round 2.

2. Four players, mostly in charge of the yellow river, had a 'good to acceptable' performance in round 1, purchased a second forecast set for 10,000 tokens or higher, and performed well in round 2.

3. Three participants, all green river players, had a 'good to acceptable' performance in round 1, bought a second forecast set for 6,000 to 8,000 tokens, but performed badly in round 2.

The winners of the game, six players in total, finished round 2 with 8,000 or 12,000 tokens in their purse. Half of these

participants were assigned the green river and the other half the blue river. Apart from one participant, all had received a biased forecast set in the first round. Most participants had a 'good' performance in the first round, did not purchase any forecast set and had an 'acceptable' performance in the second round. Their performance in round 2 did not lead to large money losses, as





it did for yellow river participants, which can be explained by the fact that they did not have so many flood events in this round (see Table 1).

The average 'avoided cost', the average bid for a second forecast set and the average 'new bid' are presented in Table 5 for each river. By comparing the 'avoided cost' with the average bid for each river, it is noticeable that the average bid was larger than the 'avoided cost' of each river, by less than 1,000 tokens for the yellow river and slightly more than 1,000 tokens for the other two rivers. Participants therefore paid on average 1,000 tokens more for their second forecast set than the benefit, in terms of tokens spared in the second round, that they made from having this forecast set. This could explain why none of the winners of the game had a forecast set in the second round. From the average 'new bid', it is evident that participants would have liked to pay less on average than what they originally paid for their second forecast set. For all the rivers, the average 'new bid' is closer to the 'avoided cost' than the average bid of participants during the auction. For the yellow river, the average 'new bid' is even lower than the average 'avoided cost'.

## 4   Discussion and conclusions

The challenging question "How much are you prepared to pay for a forecast?" was the title of a game designed to be played at several conferences and workshops in 2015. This game was set up as an experiment, of which the central element was an auction to sell a limited number of forecast sets. The aim of this experiment was to study the factors determining the decision-makers' willingness-to-pay (WTP) for a forecast set when confronted with the risk of flooding and after having previously experienced the decision-making problem in a first round. A total of 129 worksheets were collected, which analysis highlighted interesting results. However, being set up as a game, this study presents some limitations.

As mentioned by Breidert et al. (2006), a source of bias in most laboratory experiments is their artificial setup. Indeed, under such circumstances, participants are not directly affected by their decisions as they neither use their own money, nor is the risk a real one. This might lead them to make decisions which they would normally not make in real life or in operational forecasting contexts. Moreover, in our game, the costs given to both flood protection and flood damages were not chosen to represent the real costs that one encounters in real environments. First, real costs in integrated flood forecasting and protection systems are difficult to assess, given the complexity of flood protection and its consequences. Secondly, the external imposed conditions for playing our game (i.e. the fact that we wanted to play it during oral talks in conferences, workshops or teaching classes, with expected eclectic audiences of variable sizes, having a limited amount of time, and using paper worksheets to be collected at the end of the game for the analysis) were not ideal to handle any controversy on the realism (or absence of realism) of the game scenario.

Despite the inherent limitations to the setup of this game, some noteworthy points arose from the questions investigated throughout this paper. It was clear during the game that most participants had used the probabilistic forecasts they were given at the beginning of the game to help them in their decisions. This was an important issue in our game since it was an essential condition to then be able to evaluate how the participants were using their forecasts and to understand the links between the way they perceived the quality of their forecasts and the way they rated their performance at the end of a round. There was





evidence that participants were mostly using the $50^{th}$ percentile of the forecast distributions, but, interestingly, the median only could not explain all the decisions made. Other aspects of the game might have also shaped the participants' use of the information, such as the observed discovery, during the first round, of the forecast set bias (i.e. two out of three forecast sets were purposely biased for round 1). This was also mentioned by some participants at the end of some applications of the game, who said that the fact of noticing the presence of a bias (or suspecting it, since they were not told beforehand that the forecasts were biased) led them to adjust the way they were using the information. Interestingly, in the analysis of the worksheets, there was an indication that the players had, however, different tolerances to the different biases. Indeed, a lower tolerance for under-predictive forecasts than for over-predictive forecasts was identified. There was additionally evidence that, in some cases, some participants with unbiased forecasts did not use their forecasts (when considering the $50^{th}$ percentile as key forecast information). The analysis suggested that the players' risk perception might have been a reason for this. When looking at the initial river level and the proximity of the forecast final level to the flood threshold in these particular cases, there was an indication that both equally influenced the decisions made. A similar finding was reported by Kirchhoff et al. (2013) through a case study in America, where it was found out that the perception of a risk was a motivational driver of water manager's use of climate information. There is a constant effort from forecasters to produce and provide state-of-the-art probabilistic forecasts to their users. However, it was seen here that even participants with unbiased forecasts did not always use them. This is an indication that further work needs to be done on fostering communication between forecasters and users, to promote enhanced use of the information contained in probabilistic forecasts.

Many papers have shown, through different approaches, the expected benefits of probabilistic forecasts versus deterministic forecasts for flood warning (e.g., Buizza (2008); Verkade and Werner (2011); Pappenberger et al. (2015); Ramos et al. (2013)). However, many challenges still exist in the operational use of probabilistic forecasting systems and the optimisation of decision-making. This paper is a contribution to improve our understanding of the way the benefits of probabilistic forecasts are perceived by the decision-makers. It proposes to investigate it under a different perspective, by allowing, through a game experiment, decision-makers to bid for a probabilistic forecast set during an auction. The auction was used in this paper as an attempt to characterise and understand the participants' WTP for a probabilistic forecast in the specific flood protection risk-based experiment designed for this purpose. Our results indicate that the WTP displays dependencies on various aspects. The bids showed a positive relation with the money that was available to participants to make their bids. Nonetheless, this was mainly a factor for participants who had little money left in their purses at the time of the auction. The participants' perceived forecast quality of their first forecast set was also a factor influencing their WTP for another forecast set. Players who had played the first round with biased forecasts were less prone to disburse money for another forecast set for the second round. Previous experience with a forecast set seems therefore to also play a role in the WTP for forecast information to help in making decisions. There was moreover an indication that the flood frequency of the river in the first round might have influenced the WTP for a forecast set for the second round. Some players who were in charge of the yellow river, with only one flood event in the first round (i.e. low flood risk), did not consider beneficial the purchase of a forecast set for the second round. However, it must be noted that all above mentioned factors (i.e. money left in purse at the time of the auction, quality of the previous forecast set and flood frequency of the river) are interrelated factors: high flood frequency and biased forecasts can





lead to less money in purse, as players have to pay for flood protection more often, and, consequently, to a lower WTP for a forecast set when playing the second round. In order to investigate the relative influence of each factor separately in the context of the game presented here, a higher number of cases in each combination of factors would be necessary Seifert et al. (2013) have shown that "the demand for flood insurance is strongly positively related to individual risk perceptions". From our game

results, there is an indication that risk perception may also interrelate with other influencing factors mentioned previously to lead to different WTP behaviours. Some participants with a more low-risk tolerance (risk-averse behaviour) reported that they did not purchase a forecast set for the second round because they had enough tokens left in purse to protect all the time during the second round. On the other hand, a preference for "gambling" (risk-seeking behaviour) was considered a reason for not buying a forecast set by other few participants.

These results show that the perceived benefit of probabilistic forecasts as a support of decision-making in a risk-based context is multifaceted, and varies not only with the quality of the information and its understanding, but also with the relevance and the risk-tolerance of the user. This further demonstrates that more work is needed not solely to provide guidance on the use of probabilistic information for decision-making, but also to develop efficient ways to communicate the actual relevance and evaluate the long-term economic benefits of probabilistic forecasts for improved decisions in various applications of prob-

abilistic forecasting systems within the water sector. This could additionally provide insights into bridging the gap between the theoretical or expected benefit of probabilistic forecasts in a risk-based decision-making environment and the perceived benefits by key users.

This paper was an attempt to depict behaviours in the flood forecasting and protection decision-making context. Although game experiments offer a flexible assessment framework, comparatively to real operational configurations, it is however ex-

tremely complex to search for general explanatory behaviours in such a context. This is partially due to the uniqueness of individuals and the interrelated factors that might influence decisions, which are both aspects that are difficult to evaluate when playing a game with a large audience. A solution to overcome this, as proposed by Crochemore et al. (2015), could be to prolong the game by incorporating a discussion with the audience, aiming at understanding the motivations hidden underneath their decisions during the game. Having more time available to apply the game would also allow playing more cases in each

round, bringing additional information to the analysis. Moreover, although this game enabled the investigation of many aspects of decision-making and the use of probabilistic forecasts in a flood protection context, its complex structure was its strength as well as its weakness. When analysing the game results, the chicken and egg situation arose. Several factors of the participants' use of the forecasts and of their WTP for a forecast set were identified, but it was not possible to measure causalities. It would therefore be interesting to carry out further work in this direction by, for instance, testing the established factors separately.

Playing more cases could have additionally helped to clarify key aspects of the game, such as the effect of the bias on the participants' use of the forecasts and on their WTP for more forecasts. Supplementary elements could as well be added to the game in order to make it more realistic or to tailor it to specific needs, such as the incorporation of the notion of humanitarian crisis after a flood event and the addition of multiple flood thresholds with their corresponding varying damage costs.



**Resources**

This version of the game is licensed under CC BY-SA 4.0 (Creative Commons public license). It is part of the activities of HEPEX (Hydrologic Ensemble Prediction Experiment) and is freely available at www.hepex.org. This game was inspired by

5   the Red Cross/Red Crescent Climate Centre game "Paying for Predictions".






**Appendix A: Example of a worksheet distributed to the game participants (here, for river blue and forecast set type 1)**

## BLUE-1

### How much are you prepared to PAY for a forecast?

Occupation (student, PhD candidate, scientist, operational hydrologist, forecaster, professor, lecturer, other):

................................

How many years of experience do you have?  □ < 5 years  □ 5 to 10 years  □ > 10 years

> **Flood protection = -2,000 tokens; flood without protection = -4,000 tokens**
>
> **Flood occurs at 90 or above**

| Round | Case | River level before rainfall (10-60) | Flood protection? | River level increment (10-80) | River level after increment | Flood? (≥ 90) | Tokens spent | Purse (20,000) |
|-------|------|------|------|------|------|------|------|------|
| 1 | 1 | | Yes □ No □ | | | Yes □ No □ | | |
| | 2 | | Yes □ No □ | | | Yes □ No □ | | |
| | 3 | | Yes □ No □ | | | Yes □ No □ | | |
| | 4 | | Yes □ No □ | | | Yes □ No □ | | |
| | 5 | | Yes □ No □ | | | Yes □ No □ | | |

- How was your forecast set in Round 1?
  □ very bad  □ quite bad  □ neither good nor bad  □ quite good  □ very good  □ I don't know
- How was your performance as a decision-maker in Round 1?
  □ very bad  □ quite bad  □ neither good nor bad  □ quite good  □ very good  □ I don't know

Round 2 ▷





**Do not forget to transfer your final Round 1 purse to Round 2 (in the brackets under 'Purse')**

| Round | Case | River level before rainfall (10-60) | Flood protection? | River level increment (10-80) | River level after increment | Flood? (≥ 90) | Tokens spent | Purse (...........) |
|-------|------|------|------|------|------|------|------|------|
| | | *Bid: ............. tokens. Did you buy a probabilistic forecast set? YES / NO* | | | | | | |
| | | *If yes, deduct the money you paid for it here:* | | | | | | |
| 2 | 1 | | Yes ☐ No ☐ | | | Yes ☐ No ☐ | | |
| | 2 | | Yes ☐ No ☐ | | | Yes ☐ No ☐ | | |
| | 3 | | Yes ☐ No ☐ | | | Yes ☐ No ☐ | | |
| | 4 | | Yes ☐ No ☐ | | | Yes ☐ No ☐ | | |
| | 5 | | Yes ☐ No ☐ | | | Yes ☐ No ☐ | | |

- How was your performance as a decision-maker in Round 2?
  ☐ very bad    ☐ quite bad    ☐ neither good nor bad    ☐ quite good    ☐ very good    ☐ I don't know

- For the people who DID NOT buy a forecast set:

  - Why didn't you buy a forecast set?
    ☐ The model did not seem reliable
    ☐ The price was too high
    ☐ Other reason (explain): ....................................................................................................

- For the people who DID buy a forecast set:

  - How was your forecast set in Round 2?
    ☐ very bad    ☐ quite bad    ☐ neither good nor bad    ☐ quite good    ☐ very good    ☐ I don't know

  - Were the forecasts worth what you paid for them? ☐ Yes    ☐ No

  - If not, how many tokens would you now pay for them? ............ tokens

**Please return this worksheet into the envelope and give it to one of the assistants before you leave.**
**Thank you for your participation!**
**We hope you enjoyed it!**



*Acknowledgements.* This project has received funding from the European Union's Horizon 2020 research and innovation programme under grant agreement No 641811. The authors would like to thank the participants of the game who very enthusiastically took part in this experiment. We would furthermore like to acknowledge L. Crochemore, A. Ficchi, C. Poncelet and P. Brigode's valuable help with the game preparation and worksheets' distribution at the beginning of the game played at EGU 2015. Finally, we would like to thank C. Bachofen and
5   everyone who tested and gave suggestions to improve the game during its development.





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



**Table 1.** Number of flood events for each round of the game and each river.

| Round | River | | |
|---|---|---|---|
| | Yellow | Green | Blue |
| 1 | 1 | 2 | 3 |
| 2 | 3 | 2 | 1 |
| Total | 4 | 4 | 4 |





**Table 2.** Contingency table for each river and forecast set type for the first round (considering the $50^{th}$ percentile, i.e., the median forecast). The numbers for a specific river-forecast set type represent, clockwise from the top left: hits (italics), false alarms (bold), correct negatives (-) and misses (regular).

| Forecast set type | River | | | | | |
| --- | --- | --- | --- | --- | --- | --- |
| | Yellow | | Green | | Blue | |
| 1 | *1* | **1** | *2* | **1** | *3* | **2** |
| | 0 | - | 0 | - | 0 | - |
| 2 | *1* | **0** | *2* | **0** | *3* | **0** |
| | 0 | - | 0 | - | 0 | - |
| 3 | *0* | **0** | *2* | **0** | *2* | **0** |
| | 1 | - | 0 | - | 1 | - |



**Table 3.** Distribution of the 129 worksheets collected for the analysis per river (yellow, green and blue) and forecast set type (1: positively bias, 2: unbiased and 3: negatively biased).

| Forecast set type | River | | | Total |
|:---:|:---:|:---:|:---:|:---:|
| | Yellow | Green | Blue | |
| 1 | 15 | 11 | 18 | 44 |
| 2 | 13 | 21 | 9 | 43 |
| 3 | 11 | 19 | 12 | 42 |
| Total | 39 | 51 | 39 | 129 |





**Table 4.** Distribution of the forty-four second forecast sets sold during the auction, per river (yellow, green and blue) and forecast set type (1: positively biased, 2: unbiased and 3: negatively biased).

| Forecast set type | River | | | Total |
|:---:|:---:|:---:|:---:|:---:|
| | Yellow | Green | Blue | |
| 1 | 5 | 2 | 6 | 13 |
| 2 | 6 | 9 | 3 | 18 |
| 3 | 3 | 10 | 0 | 13 |
| Total | 14 | 21 | 9 | 44 |





**Table 5.** Average values of 'avoided cost' for round 2, average bid for a second forecast set and average 'new bid' if forecasts were considered not worth the price originally paid. Values are in tokens and for the three different rivers.

| River | Average 'avoided cost' | Average bid | Average 'new bid' |
|---|---|---|---|
| Yellow | 7251 | 7929 | 7083 |
| Green | 5829 | 7083 | 6224 |
| Blue | 5711 | 6889 | 5875 |





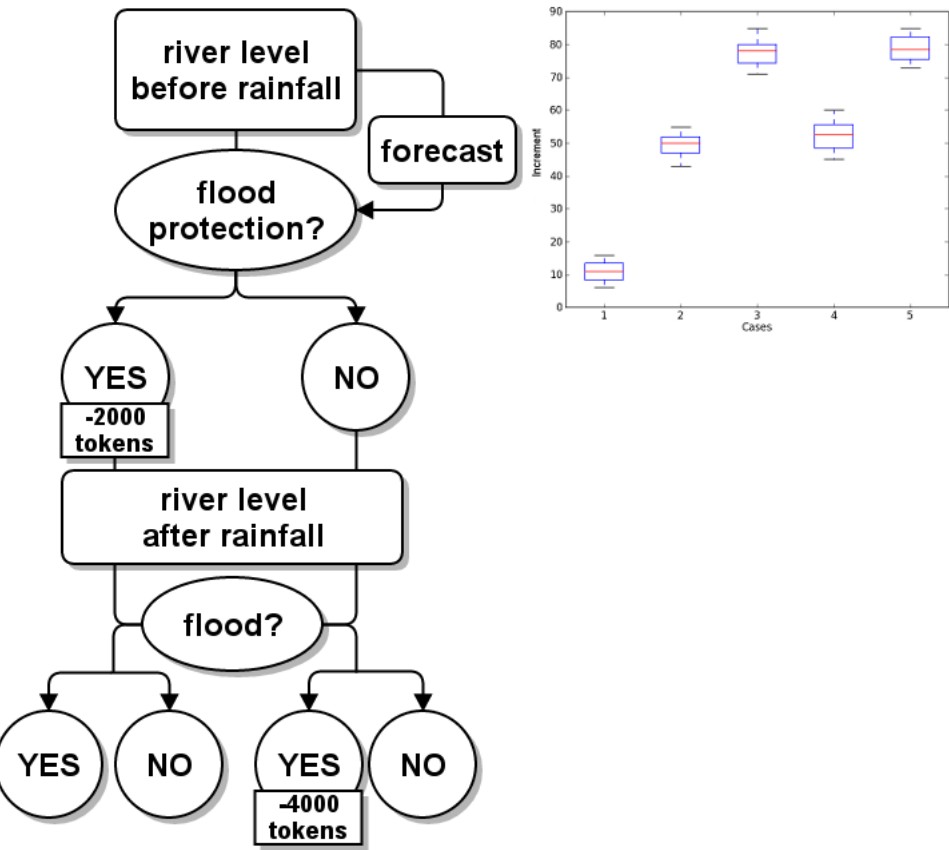

**Figure 1.** Flow diagram of the game decision problem for one case (left), and example of a forecast set, as given to the participants during the game (right). Boxplots show the $5^{th}$, $25^{th}$, $50^{th}$, $75^{th}$ and $95^{th}$ percentiles of the forecast river level increment after rainfall.





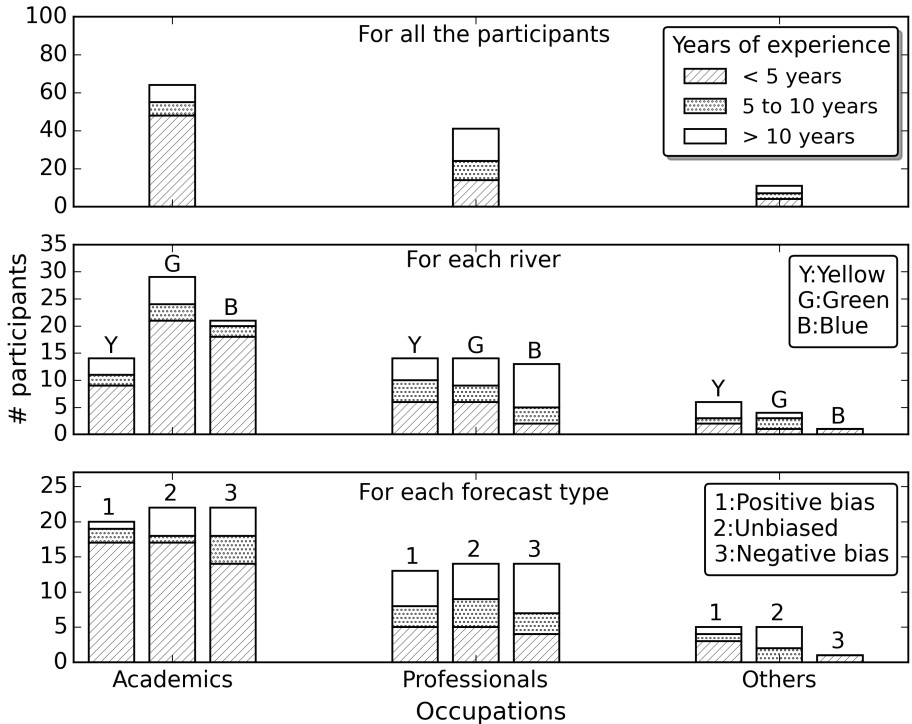

**Figure 2.** Number of participants according to occupation and years of experience. The categories of occupations are: academics (postdoctoral researchers, PhDs, research scientists, lecturers, professors and students), professionals (forecasters, operational hydrologists, scientists, engineers and consultants) and others. Top: overall participants distribution; middle: distribution according to their river; bottom: distribution according to the forecast quality types (1: positively biased, 2: unbiased and 3: negatively biased).





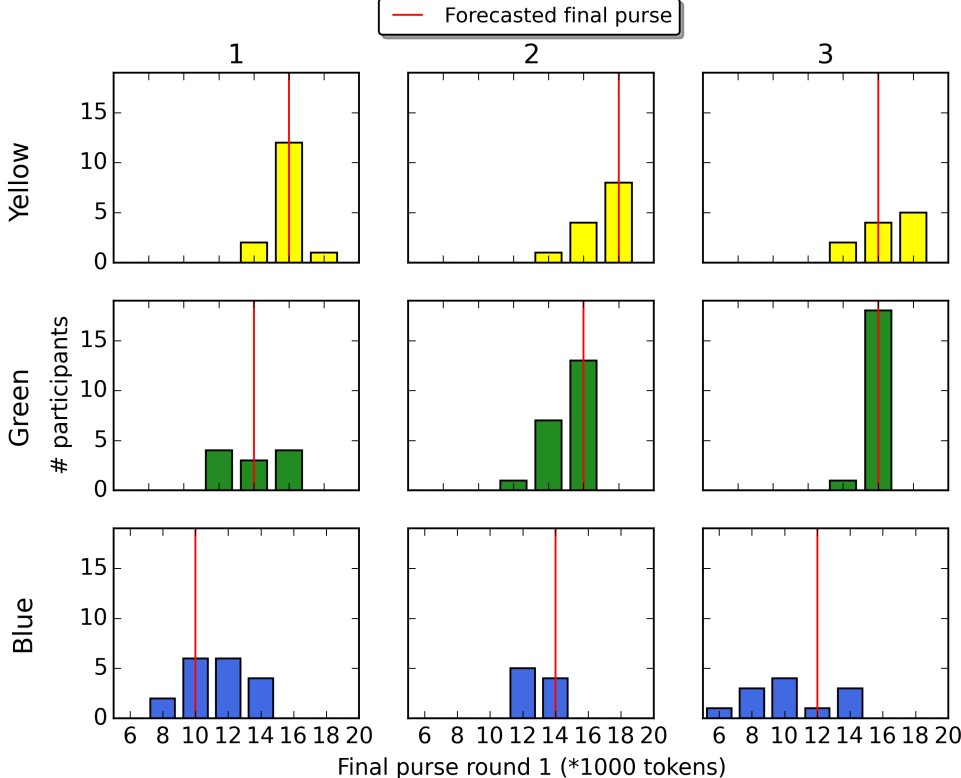

**Figure 3.** Participants' round 1 final purses for each river (from the top to the bottom row: the yellow, the green and the blue river) and for each forecast set type (from the leftmost to the rightmost column: type 1, 2 and 3). The red lines show the final purses that the participants of a given river-forecast set type group would have gotten if they had followed the median of their forecasts for all the five cases of the first round.





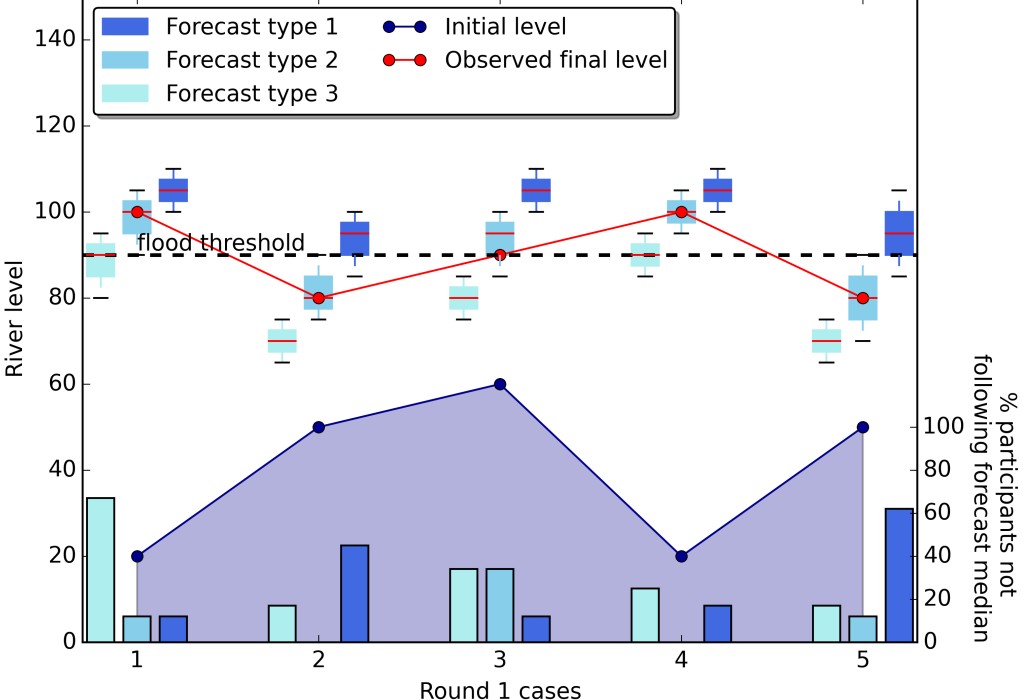

**Figure 4.** Observed initial and final river levels for the blue river for each case of the first round. The boxplots show the forecast final river levels by each forecast set type. The bars display the percentages of participants whose decisions did not correspond to what their forecast median indicated.




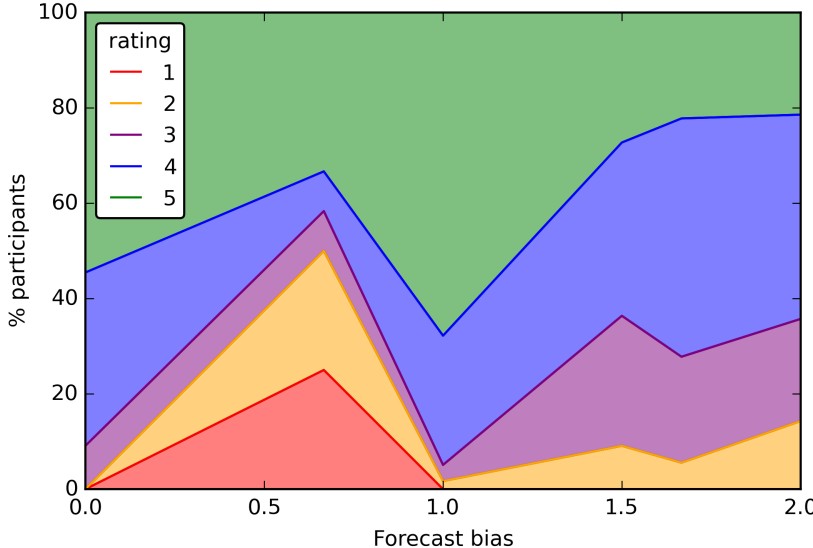

**Figure 5.** Cumulative percentages of participants that rated their forecast quality as 'very bad' (1), 'quite bad' (2), 'neither good nor bad' (3), 'quite good' (4) and 'very good' (5), as a function of the forecast bias ('true' forecast quality; Eq. (1)) in round 1. A bias equal to 1 indicates perfect forecasts, a bias less than (superior to) 1 indicates under (over) prediction.





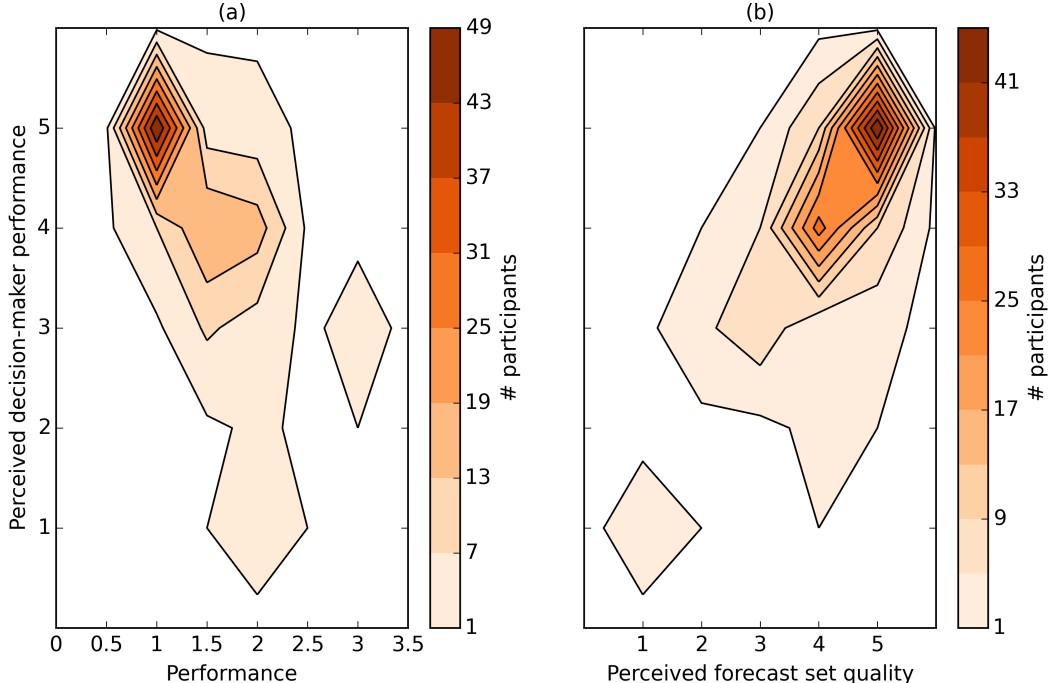

**Figure 6.** Number of participants having rated their performance as a decision-maker as 'very bad' (1), 'quite bad' (2), 'neither good nor bad' (3), 'quite good' (4) and 'very good' (5) in round 1, as a function of: (a) their 'true' performance (calculated from Eq. (2)), and (b) their rating (same scale from 1 to 5 as above) of their forecast set quality, or in other words their perceived forecast set quality. A performance value of one denotes a 'true' performance equal to the optimal performance (Eq. (3)). The larger the performance value, the more distant from the optimal the decisions were during round 1.





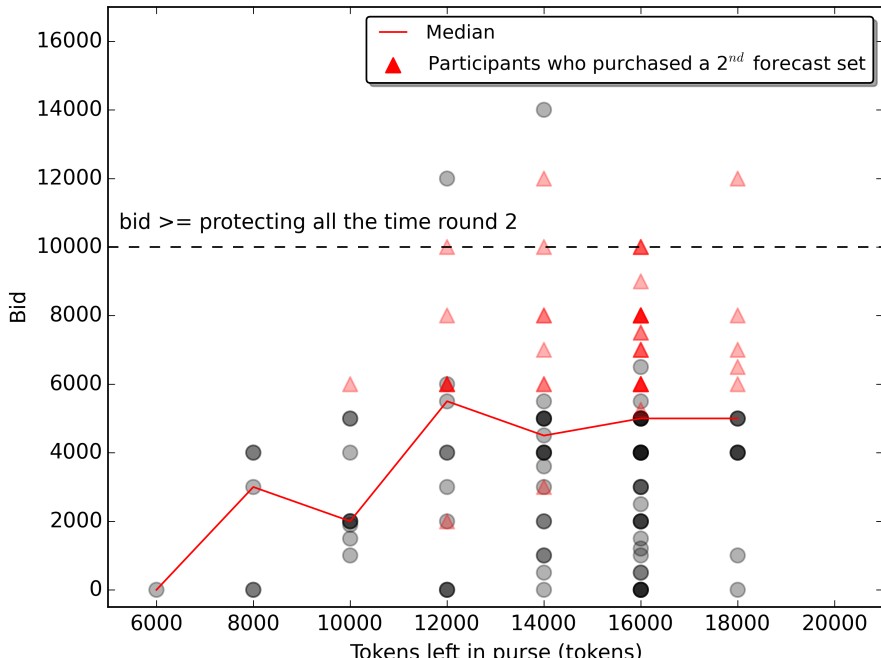

**Figure 7.** Bids declared by participants to purchase a forecast set for round 2, as a function of the amount of tokens they had left in their purse at the end of round 1. The colour of the points indicates the number of participants represented by this specific point. The darker the colour, the larger the number of participants.




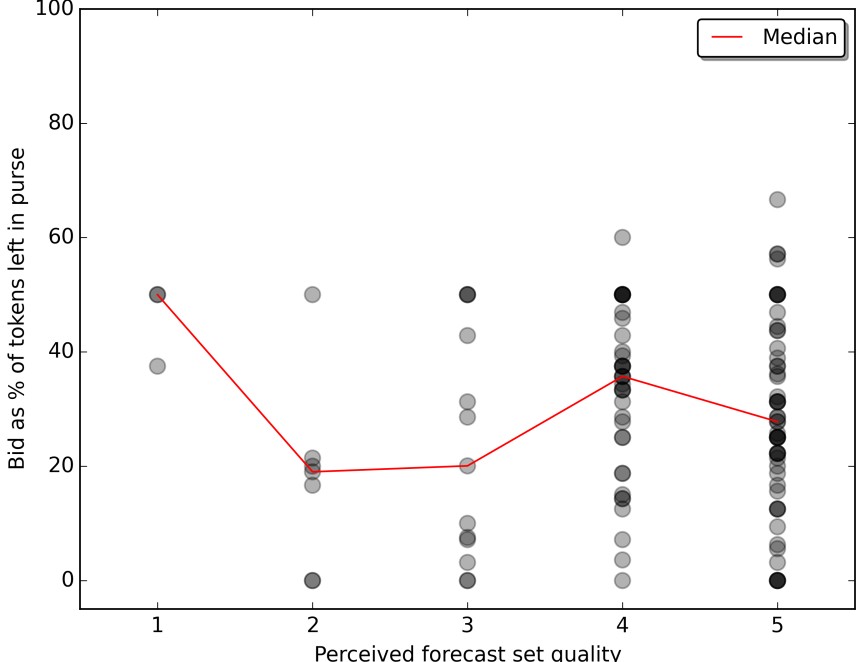

**Figure 8.** Participants' % bids, bids expressed as a percentage of the tokens participants had left in purse at the time of the auction, as a function of the rating they gave to their forecast set quality in round 1 (from 1, 'very bad', to 5, 'very good'; see Appendix). The colour of the points indicates the number of participants represented by this specific point. The darker the colour, the larger the number of participants.