# Peer review of "Willingness-to-pay for a probabilistic flood forecast: a risk-based decision-making game"

_Hydrology and Earth System Sciences, 2016_

## Referee Comment (RC1) · Anonymous Referee #1 · 19 Feb 2016

GENERAL COMMENT: This manuscript brings interesting insights about the value of flood forecasts for decision making. As well stated by the authors, the field of probabilistic flood forecast has evolved a lot in the past years. However, there is still a gap on how to interpret the probabilistic flood forecasts and use it for decision making, and how valuable are it is for decision makers. SO this paper reports an interesting experiment that brings new reflections on both the experiment's participants and the reader of NHESS/HESS. It was interesting to read it, so it is generally well written. So I'll be pleased to see it published in NHESS/HESS. These are minor questions/comments that hopefully will help the authors to improve the presentation of their manuscript:

-Experiment Setup: How conclusions could be impacted by the choice of the parameters for the experiment? e.g. protection cost or damage costs? How river levels and increments were sampled??? What about the uncertainty of forecast increments? How

such choice could impact conclusions?

-Page 5.Line 10 Include a figure (or a new panel on fig. 1) showing a diagram with the sequences of rounds, auction, etc., to summarize section 2 (experiment setup).

-Page 10 Line 15. Figure 2. How this distribution compare to the distribution of people actually making the decisions?

-Page 11, Lines 5-22. Figure 3. It is really not clear from this figure 3 if participants changed strategy during the 5 cases. It would be easier to ask this question to the participants in the form.

-Page 11, Lines 20-25. Figure 3. It seems that participants also used information on the other percentiles different from median. For example, in case 4 for type 2, 5th and 95th percentiles indicate flood, so all participants chose the same and correct action. On the other hand, in cases where 5th and 95th percentiles fall above and below the flood threshold, more people did not follow the median (case 1 type 1, case 2 type 3, case 3 type 2, case 5 type 3).

-Page 12, Line 5, Figure 6b This is very interesting. Participants attribute good decision making performance with good forecast quality, but they forget that their personal strategy adopted for decision making plays a major role, as there are several ways to interpret and use the probabilistic forecasts. What is the implication in the real world? Decision makers will tend to blame (thank) forecast providers for their wrong (good) decisions?

-How the flood frequency observed in Round 1 impacted the WLP? Why participants from Green river are more WLP?

-Section 3.5: The analyses show that participants using forecasts had better performance in Round 2, however, participants with more money and better performance in Round 1 were willing to pay more for the forecasts. Consequently, participants with better performance in Round 1 ended buying forecasts and having better performance in

Round 2. How the skill of the participants could impact the conclusion that "Decisions are better when they are made with the help of unbiased forecasts, comparatively to having no forecasts at all"

-Section 3.6. : Show % values, table or figure for these results. It may help the reader.

-Is a biased forecast better than no forecast? Can you access that from this experiment?

---

## Referee Comment (RC2) · Anonymous Referee #2 · 16 Mar 2016

The authors present the evaluation of a flood protection game for users of probabilistic hydrological forecasts. The aim of the game is to spend as little money as necessary for protecting a city from flooding. The participants were divided into 9 groups differing in flood frequency and forecast bias. The nine groups have 9 to 21 members/participant. Due to this limited dataset the evaluation is more of a descriptive kind.

Nonetheless, it gives an insight into possible reasons for taking good or bad decisions. And on factors that influence the willingness to pay for a forecast.

The article is written well. After an introduction the game set up and the objectives are presented clearly. The objectives are addressed with six questions which are first described thoroughly in paragraphs 2.1.1 to 2.1.6. The answers to these questions are presented in the results section 3.1-3.6. It might be more reader friendly to combine

these sections and take the question-answer pairs one after another.

Specific Comments:

P4L16-19: Did the participants know from the beginning that there are two rounds? –> According to P6L7 they didn't.

P4L27: Total number of flood events is kept equal in order to give equal chances to all participants to win.

- This statement doesn't hold. Participants surely did not have equal chances to win the game. This depended on the forecast set type and the river they were given.

- For the aim of the study it is not required that every participant has the same chance to win, but this statement is wrong. –> 2.1.1. P5L3

P4L29-30: the number of flood events was different for every river, but not for every round –> the Green River has the same number of events in round 1 and 2, whereas it changes for Yellow River and Blue River (which again influences their chances to win).

P6L7: So, at the beginning, the participants didn't know that there were two rounds to play?

P6L26-27: How did the participants that purchased a forecast perceive the quality of the forecasts compared to round 1? Did participants that had biased forecast sets in round 1 notice the better performance of the forecasts in round 2?

Figure 3: - Yellow and Green River –> the ticks do not match the x-axis labels! - It would be nice if the plot was arranged according to table 3 - Change "forecasted final purse" to "final purse if following median forecast" - Labels of the "columns" could be changed to "pos. biased", "unbiased", "neg. biased" (same could be done for fig. 4)

QA 1 (2.2.1/3.1):

Fig 4: please change order of the bars such that forecast type 1 is first in line.
Equal chances ...: - the disadvantage for participants with positive bias is smaller if the observed value is above/equals the flood threshold. Thus for the blue river, which experiences three floods in five cases (1st round), the participants with forecastset type 1 are expected to perform better than those with forecast set type 3.

QA2 (2.2.2/3.2): You state that the percentage of negative perceptions of the quality of the forecasts increases with increasing or decreasing forecast bias. This seems to be quite consistent for positively biased forecasts, but how do you explain that the distribution of the ratings from the participants with the most negative bias (0) looks almost the same as from the participants with unbiased forecasts? –> Fig.5

QA3 (2.2.3/3.3): At a first glance Figure 6 looks completely fine. However, there were five levels of perceived performances (very bad to very good) the participants could choose from. So the graph should not show perceived performances higher than five or lover than one. You could change the graph to a simple scatterplot and choose the point size proportional to the number of participants that fall onto a specific perceived-actual-performance combination. (Same for Figure 6 b).

QA4 (2.2.4/3.4):

P13L3-4: It would be more straight forward if you just stated the average percentage the participants were willing to spend from the tokens left in their purse.

P13L4-5: ... 48+32+21=101 ... round to 20 % for blue river. However, it would be more informative what percentage of river group members purchased a forecast for the second round → 36% of the yellow river group, 41% of green rivers, 23% of blue rivers purchased forecasts for the second round. And also for the forecast set type groups → pos. biased 30%, unbiased 42% and neg. biased 31%.

QA6 (2.2.6/3.6):

P15L18-19: Could you give the average final purse for the two groups "with and without forecast in the second round" separately.

Table 5: you could do that additionally by forecast set type

Discussion/Conclusion:

P18L9: gambling was considered a reason for not buying a forecast set by other few participants. → according to P14L22 this was just one participant.

P18L12-15: "This further demonstrates that more work is needed not solely to provide guidance on the use of probabilistic information for decision-making, ….."

Comment: It is questionable if the setup of the game is not to some extent counterproductive and doesn't help to improve this. The winners of the games had mostly biased forecasts in the first round and no forecast in the second round . . .

If the game should have an educational merit, shouldn't the game then be set up in a way so that people who have no bias in the forecasts of the first round and who purchase a forecast in the second round have better chances to win?

Minor Comments:

P2L3: remove dashes

P3L15: remove "one"

P7L8: his purse

P11L17: . . . the lowest percentage of participants not following the median forecast are for the unbiased forecast set type 2.

P13L16: . . . (the river that experienced most floods in round 1 and for which players thus ended the first round with on average the lowest amount of tokens left in their purse)

P16L5-6: . . . than the "avoided cost" of each river. On average participants paid 1000 tokens more . . .

P18L3: . . . necessary. Seifert . . .

general: it would be easier for the reader if you used the terms "neg. biased forecast", "unbiased forecast" and "pos. biased forecast" instead of the terms "forecast set type 1-3" in the text, tables and graphs.

———————————————————

---

## Author Response (AR1)

**Response to first referee**

We would like to thank the reviewer for the positive and constructive remarks. We provide here below a response to the main points discussed:

- Experiment Setup:
  - How conclusions could be impacted by the choice of the parameters for the experiment? e.g. protection cost or damage costs? This is a good point. We have mentioned a possible impact of these choices in the results section (P12L9-11). When designing the game, we have tried to balance initial money in purse, costs and number of rounds in a way that players could focus more on using the information provided than on money left in hand. However, we had expected that the willingness to pay (WTP) for a service would depend on how much money you have available. That is why we have examined the bids as a function of the money left in hand after round 1. The results are presented in Section 3.4 (page 12). They show that, in our configuration, the money left in hand did not influence significantly the WTP (P13L23-24). We believe that if the damage costs were higher, for instance, participants would probably want to protect all the time, which would make forecasts rather useless (i.e., whatever is forecast to happen, participants would pay for protection anyway). We did not want to have this situation, as we would not know how they were using (or willing to use) their forecasts. It would be interesting to test the influence of the many parameters of the setup, but then we would need to make constant some other parameters and play the game differently. For instance, without different qualities of the forecasts or no differences in flood events among rivers. This could be interesting for further developments and we will mention it in the conclusion section, where we have also mentioned some other limitations of the setup (P16L19-28).
  - How river levels and increments were sampled??? What about the uncertainty of forecast increments? How such choice could impact conclusions? River levels and increments were set in a way that the number of floods was kept equal for each river, and river level values (of initial river level and river level increment) were randomly generated. This is mentioned in the experiment setup section (P4L27-30). As for the forecast increments, it was not the uncertainty (here expressed as interquantile ranges of the boxplots representing the forecasts) that differed among the different forecast sets, but the position of the observations inside the forecast distribution (which made the forecast underpredict or overpredict the observation). This was explained in the round 1 setup section (P5L16-19), but we will make it more clear in the revised version of the paper. The impact on conclusions comes therefore from the different quality (or perceived quality) of the forecasts on the decisions made. This is an issue explored in the results presented in Section 3.1 (P10). We could expect that using forecasts with different levels of uncertainty would make participants spend more money on protection when forecast increments are too uncertain (large boxplots) and take more risks (by not protecting) when forecasts are sharper. This could be an option for another setup of the game (i.e., instead of using forecasts of different accuracy).

- Page 5.Line 10 Include a figure (or a new panel on fig. 1) showing a diagram with the sequences of rounds, auction, etc., to summarize section 2 (experiment setup). This is a good idea, we believe that it will make the game structure clearer to the readers. Therefore, a diagram summarising the setup of the experiment will be added as a panel on Figure 1.

- Page 10 Line 15. Figure 2. How this distribution compare to the distribution of people actually making the decisions? This is a difficult question to assess. We have not included a question in the worksheet about it and have not discussed this point after the game. It would be certainly interesting to know the percentage of each category that had actual decision-making duties in their work. This is an issue that we can add in the discussion and conclusions section for further improvements of the game.

- Page 11, Lines 5-22. Figure 3. It is really not clear from this figure 3 if participants changed strategy during the 5 cases. It would be easier to ask this question to the participants in the form. This is a good point, but one that raises several challenges. In fact, participants were not aware that there could have been a bias in their forecasts. It was important that they did not know that in advance. If we had a direct question on the worksheet (which they have in hand from the beginning of the game), they would have discovered this feature (or be suspicious) before starting to play. Also, if we had put a question asking if they had looked at the median, or the upper or the lower quantiles, we would already be suggesting in advance that they should be looking at one of these. We did not want to influence their strategy. It is a difficult issue when designing a game experiment: how much do we present before they start playing and how much do we leave for the participants to decide (i.e., to play freely)? We have opted to leave it free to participants and then analyse the responses under some hypotheses and in terms of "could the participants have discovered the bias in their forecasts?" This was motivated by the fact that several participants came to talk to us after the application of the game saying that they had seen that their forecasts were biased. Therefore, this was our main driver in the analysis of "possible influencing factors" in Section 3.1.

- Page 11, Lines 20-25. Figure 3. It seems that participants also used information on the other percentiles different from median. For example, in case 4 for type 2, 5th and 95th percentiles indicate flood, so all participants chose the same and correct action. On the other hand, in cases where 5th and 95th percentiles fall above and below the flood threshold, more people did not follow the median (case 1 type 1, case 2 type 3, case 3 type 2, case 5 type 3). This is a very good point and it will be added to the results (Section 3.1).

- Page 12, Line 5, Figure 6b. This is very interesting. Participants attribute good decision making performance with good forecast quality, but they forget that their personal strategy adopted for decision making plays a major role, as there are several ways to interpret and use the probabilistic forecasts. What is the implication in the real world? Decision makers will tend to blame (thank) forecast providers for their wrong (good) decisions? This is a very good point which will be raised in the Discussion Section of the paper. It was mentioned in a HEPEX post "On the economic value of hydrological ensemble forecasts". ([http://hepex.irstea.fr/economic-value-of-hydrological-ensemble-forecasts/](http://hepex.irstea.fr/economic-value-of-hydrological-ensemble-forecasts/)). "*We once asked a decision-maker responsible for deciding on whether or not to open a control gate of a dam, and of how many meters it should be opened in case the decision was to open it, how he knew afterwards that his decision was the 'best decision'. The answer we got was*

*(not* ipsis verbis*): 'It is the best decision: we take the best decision given the forecasts we receive and other complementary information we have on the situation. If the result is not good, the problem is not in the decision, but in the forecasts, which were not good'."*

- **How the flood frequency observed in Round 1 impacted the WTP? Why participants from Green river are more WTP?** The effect of the flood frequency of round 1 on participants WTP for another forecast set was briefly mentioned on P14L18-19 and in the conclusions (P17L31-33). However, nothing was said about the higher percentage of green river participants buying a second forecast set. We believe that it is a combination of the flood frequency (not as low as for the yellow river, which made it more relevant for green river participants to buy a second forecast set) and of money left in purse (on average, not as low as the blue river's participants). This will be added to the analysis section of the paper.

- **Section 3.5: The analyses show that participants using forecasts had better performance in Round 2, however, participants with more money and better performance in Round 1 were willing to pay more for the forecasts. Consequently, participants with better performance in Round 1 ended buying forecasts and having better performance in Round 2. How the skill of the participants could impact the conclusion that "Decisions are better when they are made with the help of unbiased forecasts, comparatively to having no forecasts at all".** We understand that the general question can be unfolded into the following questions: does the conclusions pertain only to the "good performing" participants? Is it true also for the "bad performing" participants? Do you need already to be a good decision maker to benefit from having forecasts in hand? Or do bad decision makers also improve their decisions when having forecasts? In order to investigate this issue, we first looked at the number of participants with a bad performance in round 1 and who had a forecast in round 2: all of these participants had a good performance in round 2. This is an indication that even when participants had a bad performance in round 1, when they had a forecast set in round 2, they all had a good performance in the second round. We then looked at the number of participants with a good performance in round 1 and who had no forecast in round 2. 57 out of 59 had a bad performance in round 2. This is also an indication that even if participants had a good performance in round 1, if they had no forecast set in round 2, they mostly had a bad performance. These observations will be added to the analysis in Section 3.5 in the revised version of the paper.

- **Section 3.6. : Show % values, table or figure for these results. It may help the reader.** We will add a table to back up the text about the winning and losing strategies.

- **Is a biased forecast better than no forecast? Can you access that from this experiment?** This is an interesting question: are forecasts, even if biased, useful for decision-making, comparatively to having no forecasts at all? Our experiment suggests that some participants adjusted their biased forecasts to make their decision. This can be an indication that they were useful somehow. However, our experiment setup does not allow drawing general conclusions. We cannot directly compare the performance of participants with biased forecasts in round 1 with the performance of participants without any forecasts in round 2, since the situations were not the same in both rounds (i.e., initial river levels, river level increments, money in purse, etc). In order to fully investigate this particular issue we would need to have an experiment where we play the same cases with two groups: one having no forecasts and another having biased forecasts. It is, in fact, interesting to note that when we

setup a game experiment and analyse the results, several other possibilities open up for new variants of the experiment and further investigations. For us, this shows that there are still several open opportunities to enhance our understanding on how forecasts can be better used to inform decision-making.

**Response to second referee**

We would like to thank the reviewer for the positive and constructive remarks. We provide here below our answers to the points discussed:

*It might be more reader friendly to combine these sections and take the question-answer pairs one after another.* We have thought about it when writing the paper but some feedback from the first readers found that it would be better to separate methodology from results.

Specific comments:

- *P4L16-19: Did the participants know from the beginning that there are two rounds? → According to P6L7 they didn't. And P6L7: So, at the beginning, the participants didn't know that there were two rounds to play?* P6L7 indeed suggests that the participants were not aware that 2 rounds would be played in total. However, participants were given a worksheet at the beginning of the game which contained a table for both round 1 and round 2. Moreover, they were told that 2 rounds of 5 cases each would be played during the game introduction. This will be clarified in the revised version of the paper.

- *P4L27: Total number of flood events is kept equal in order to give equal chances to all participants to win. This statement doesn't hold. Participants surely did not have equal chances to win the game. This depended on the forecast set type and the river they were given. For the aim of the study it is not required that every participant has the same chance to win, but this statement is wrong. → 2.1.1. P5L3.* This is a good comment and the reviewer is right for pointing it out. An equal total number of floods is not a needed criteria for this specific experiment and does not give equal chances to participants to win the game, given the other influencing factors. This sentence will be removed from the revised paper.

- *P4L29-30: the number of flood events was different for every river, but not for every round → the Green River has the same number of events in round 1 and 2, whereas it changes for Yellow River and Blue River (which again influences their chances to win).* These flood frequencies were chosen in order to: 1) explore the influence of different flood frequencies in round 1 on the participants WTP for a second forecast set, 2) investigate the change (yellow and green rivers) or no change (blue river) in flood frequency between round 1 and round 2 on the participants' strategies throughout the game. This was presented in Section 3.5 (P15L6-13). We will clarify those motivations in the revised version of the paper.

- *P6L26-27: How did the participants that purchased a forecast perceive the quality of the forecasts compared to round 1? Did participants that had biased forecast sets in round 1 notice the better performance of the forecasts in round 2?* This is an interesting point. It was explored during the analysis of the game results but only partially stated in the paper (see Section 3.5, P15L14-15). Among the 44 participants who bought a forecast in round 2 and perceived it of "good quality", 18 had played round 1 with a biased forecast set, and 19 with an unbiased forecast set. The 3 remaining participants stated that their forecasts for the second round were 'neither good nor bad' or 'quite bad'. These 3 participants all had biased forecasts in the first round and their behaviour during round 2 suggested that they might have been influenced by the bias in their forecasts during round 1. These additional results will be added to the paper (Section 3.5).

- *Figure 3: - Yellow and Green River –> the ticks do not match the x-axis labels! - It would be nice if the plot was arranged according to table 3 - Change "forecasted final purse" to "final purse if following median forecast" - Labels of the "columns" could be changed to "pos. biased", "unbiased", "neg. biased" (same could be done for fig. 4).* We thank the referee for the suggestions and will change the figures accordingly.

- QA 1 (2.2.1/3.1):
  - Fig 4: please change order of the bars such that forecast type 1 is first in line. This will indeed make the figure easier to read. We will change as suggested.
  - Equal chances…: - the disadvantage for participants with positive bias is smaller if the observed value is above/equals the flood threshold. Thus for the blue river, which experiences three floods in five cases (1st round), the participants with forecast set type 1 are expected to perform better than those with forecast set type 3. This is a good point, and it will therefore be added to the paper. We indeed expected an influence of the game setup on the participants' behaviours and on their performances. The costs chosen for this experiment is an example of those factors. It was mentioned on P12L7-11 that, as the damage cost was set to twice the protection cost, this might have influenced the participants' tolerance to misses and false alarms.
- QA2 (2.2.2/3.2): You state that the percentage of negative perceptions of the quality of the forecasts increases with increasing or decreasing forecast bias. This seems to be quite consistent for positively biased forecasts, but how do you explain that the distribution of the ratings from the participants with the most negative bias (0) looks almost the same as from the participants with unbiased forecasts? → Fig.5. Participants with a bias of 0 belonged to the yellow river and had negatively biased forecasts in round 1. There was only one flood for river yellow, which occurred at the end of round 1. The negatively biased forecasts for river yellow thus missed this flood. During the analysis of the results, it was observed that only about 25% of the yellow river participants given the negatively biased forecasts did not protect for this flood. This could be because the participants had time to learn about their forecasts' quality until the occurrence of the flood, during case 5 of round 1. This low number of participants who actually suffered from their negative bias and the presence of only 1 miss out of the 5 cases of round 1 could explain the good rating of their forecasts by those participants. This will be made clearer in the text of the revised version.
- QA3 (2.2.3/3.3): At a first glance Figure 6 looks completely fine. However, there were five levels of perceived performances (very bad to very good) the participants could choose from. So the graph should not show perceived performances higher than five or lover than one. You could change the graph to a simple scatterplot and choose the point size proportional to the number of participants that fall onto a specific perceived actual-performance combination. (Same for Figure 6 b). This is true and a good suggestion. The figure will be modified accordingly.
- QA4 (2.2.4/3.4):
  - P13L3-4: It would be more straightforward if you just stated the average percentage the participants were willing to spend from the tokens left in their purse. We think that both, the average percentage and the actual amount, are needed for the sake of clarity.
  - P14L4-5: … 48+32+21=101 … round to 20 % for blue river. Yes, good point! However, it would be more informative what percentage of river group members purchased a forecast for the second round → 36% of the yellow river group, 41% of green rivers, 23% of blue rivers purchased forecasts for the second round. And also for the forecast set type groups → pos. biased 30%, unbiased 42% and neg. biased 31%. The percentages within each group will be added to follow the suggestions of the reviewer. We also think it will be clearer for the reader.
- QA6 (2.2.6/3.6):
  - P15L18-19: Could you give the average final purse for the two groups "with and without forecast in the second round" separately. On average, the participants without a second forecast set ended the game with 3341 tokens, compared to 2778 tokens on average for participants with a second forecast set. These final average

purses are however not only a reflection of the participants' overall performances, but also of the setup of the game. If the number of cases had been larger in round 2 (not feasible due to the restricted amount of time in our case), we would possibly have seen a larger benefit of having a forecast set in round 2 and as a result, a larger final purse for participants with a second forecast set. This is why the analysis of the winning and losing strategies (presented in Section 3.6) is largely qualitative. The final purses will be mentioned in Section 3.6.

- o Table 5: you could do that additionally by forecast set type. It is a possibility, but we think this would make the results more difficult to interpret since the table refers to round 2 and the forecast set types played a stronger role in round 1. Also, the average avoided cost makes sense for the same river as it will only depend on the possession of a second forecast set or not (as the flood frequency is the same within a river). If we consider also the different forecast quality groups, the average avoided cost will depend on the possession of a second forecast set but also on the number of floods for the second round and on the distribution of the participants among the different rivers. This would stratify too much our sample and results would lose robustness.

- Discussion/Conclusion:
  - o P18L9: gambling was considered a reason for not buying a forecast set by other few participants. → According to P14L22 this was just one participant. Thank you for pointing this out. It will be corrected in the revised version of the paper.
  - o P18L12-15: "This further demonstrates that more work is needed not solely to provide guidance on the use of probabilistic information for decision-making,..." Comment: It is questionable if the setup of the game is not to some extent counterproductive and doesn't help to improve this. The winners of the games had mostly biased forecasts in the first round and no forecast in the second round … If the game should have an educational merit, shouldn't the game then be set up in a way so that people who have no bias in the forecasts of the first round and who purchase a forecast in the second round have better chances to win? By designing this experiment, we did not intend to create an educational game as such, but rather a role-play situation where some important topics in hydrological forecasting could be reflected upon and discussed within a group. The main purpose of the game was to answer the questions explored and presented in this paper, within the limits of its feasibility and results' interpretation. The game contributes to the discussion about how forecasts are used and how much one is ready to pay to have them in their decision-making situation. Much of what one can learn from the game relies on the discussions the group may have after playing it. We believe that issues on the importance of forecasts for decision-making can be raised by participants once they have been introduced to the topic with this game experiment and are more at ease to share their own perceptions and knowledge.

Minor Comments: The required corrections will be made for the following comments.

- P2L3: remove dashes
- P3L15: remove "one"
- P7L8: his purse
- P11L17: … the lowest percentage of participants not following the median forecast are for the unbiased forecast set type 2.
- P13L16: … (the river that experienced most floods in round 1 and for which players thus ended the first round with on average the lowest amount of tokens left in their purse)

- P16L5-6: … than the "avoided cost" of each river. On average participants paid 1000 tokens more …
- P18L3: … necessary. Seifert …
- General: it would be easier for the reader if you used the terms "neg. biased forecast", "unbiased forecast" and "pos. biased forecast" instead of the terms "forecast set type 1-3" in the text, tables and graphs.

**Relevant changes**

**2. Set up of the decision-making game**

- The sentence saying that the total number of flood events was kept equal in order to give participants equal opportunities to win the game was removed from Section 2.1. This was not a required parameter for the aim of the game, as pointed out by referee 2.
- The different numbers of floods between the two rounds of the game was mentioned, as well as the opportunity that this offered for the analysis of the game. This was included in Section 2.1 (page 4, lines 26-30), following a comment from referee 2.
- We have clarified the different types of probabilistic forecasts distributed for the first round of the game by adding the following sentence to Section 2.1.1 (page 5, lines 16-17): "The three different forecast types were obtained by varying the position of the observation inside the forecast distribution". This was added following a comment from referee 1.
- We have removed the sentence suggesting that participants did not know that there would be two rounds to play, as it was not true, previously found in Section 2.1.3. This is following a comment from referee 2.

**3. Results**

- An additional paragraph was included in Section 3.1 (pages 11-12, lines 33-2) to mention the possible use of other percentiles of the forecasts by participants during round 1. This was suggested by referee 1. It was also mentioned briefly in the "Discussion" and "Conclusions" sections.
- A paragraph was added to Section 3.2 (page 12, lines 12-19) in order to explain the very similar ratings for participants with the most negative bias (0) and participants with unbiased forecasts (Figure 5). This was an issue raised by referee 2.
- We wrote a few lines on the high WTP for participants with the green river, which were added in Section 3.4 (page 14, lines 20-23). This was done as a result of a comment from referee 1.
- The percentages of participants who bought a second forecast set among the different rivers were corrected to add up to 100% (Section 3.4, page 14, lines 18-20). Additionally, the percentage of participants with a second forecast set within each river and each forecast set type was also mentioned (Section 3.4, page 14, lines 16-20). These two points were raised by referee 2.
- We have added a paragraph in Section 3.5 (page 15, lines 21-27) arguing that you do not need to be a good decision-maker to benefit from the forecasts in hand, in the context of this experiment. This was mentioned following comments from referee 1.
- A few lines were added to Section 3.5 (page 16, lines 5-8) in order to explain further the participants' perception of their forecast set quality in round 2, compared to round 1. This was done to answer a few questions from referee 2.
- While we previously agreed to support the text about winning and losing strategies (Section 3.6) with a table (suggestion from referee 1), we have finally decided instead to include the performance values in the text, in order to make this part of the analysis slightly less qualitative and also more consistent with the rest of the paper. We felt that the table did not provide enough added value for the space required.

- The average final purses of participants without a second forecast set and with a second forecast set were added to Section 3.6 (page 16, lines 11-12; suggestion from referee 2).

**4. Discussion**

- The former "Discussion and conclusions" section was reformatted into a "Discussion" section, written into two distinct parts: "Experiment results and implications" and "Game limitations and further developments". In the first part, the results of the game and the larger questions they raise are discussed. In the second part, the limitations of the game are mentioned, and possible further developments of the experiment are suggested. This was done to improve the clarity of the discussion.

**4.1. Experiment results and implications**

- Even though we could not answer the question whether a biased forecast is better than no forecast at all from the game, we have raised this point in the "Discussion" section of the paper (page 17, lines 17-18): "This could suggest that forecasts, even biased, can still be useful for decision-making, comparatively to no forecasts at all, if users are aware of the bias and know how to consider it before taking a decision."
- The fact that biased forecasts were problematic for the users, which hints that there is an important need for probabilistic forecasts to be bias-corrected, was added (page 17, lines 21-23).
- A paragraph was added here to mention an interesting point raised by referee 1 (page 18, lines 1-7). This is the fact that participants attribute good decision-making performance with good forecast quality, but might tend to forget that their personal strategy plays a major role.
- The sentence leading to believe that multiple participants mentioned gambling as a reason for not buying a second forecast set was rephrased, as suggested by referee 2 (page 18, lines 24-25).

**4.2. Game limitations and further developments**

- We have added a paragraph in this section mentioning the impact of the choice of the parameters on the experiment's results (page 19, lines 22-26). This was suggested by referee 1.
- A few lines were added on the small sample size of this experiment, encouraging others to replicate the experiment (page 19, lines 27-29).

**5. Conclusions**

- A "Conclusions" section was included to highlight the key results of the paper.

**Tables and figures**

- A panel was added on Figure 1, showing a diagram of the experimental set up. This was added to Figure 1 after a comment from referee 1 suggested that it would make the experimental set up clearer to the readers, which we agreed to. We have also replaced the five probabilistic forecasts (as an example of the probabilistic forecast sets given to the participants for the first round) previously displayed on this figure, for a schematic of one single probabilistic forecast (displayed as a boxplot and where the boxplots' percentiles - mentioned in the paper - are apparent). This was done as there was no added value of the

five forecasts being shown and we believe that the modified image makes the figure clearer. The figure's caption was modified accordingly.

- The legend of the lowest graph of Figure 2 was changed from "positive bias", "unbiased" and "negative bias" to "pos. biased", "unbiased" and "neg. biased" respectively. This was done to keep the paper consistent.
- Figure 3 was modified slightly in order to make the changes highlighted by referee 2 (i.e. putting the rivers as columns and the forecast quality as rows to keep the consistency with the tables; naming the forecast qualities "pos. biased", "unbiased" and "neg. biased" rather than forecast set type 1, 2 and 3; and rephrasing the legend). The caption was adjusted accordingly.
- Figure 4 was modified slightly in order to make it more reader-friendly, as suggested by referee 2. This was done by changing the legend to "neg. biased", "unbiased" and "pos. biased". The caption and the corresponding text were also adjusted.
- The legend of Figure 5 was changed from rating 1 to 5, to 'very bad' to 'very good'. The caption was modified accordingly.
- Figure 6 was corrected by changing the previous contour plots to scatter plots, suggested by referee 2. The x- and y-axis of the perceived forecast set quality and perceived decision-maker performance were changed from rating 1 to 5, to 'very bad' to 'very good'. The caption was also modified.
- The x-axis of Figure 7 was changed to range between 6,000 and 18,000 tokens only. It previously showed 20,000 tokens as well, for which there is no data. The caption was also made clearer.
- The x-axis of Figure 8 was changed from rating 1 to 5, to 'very bad' to 'very good'. The caption was also made clearer.

**General comments**

- One of the co-authors' name (Erin Coughlan de Perez) and their affiliations were rectified.
- While we previously agreed to the comment of referee 2 on the blue river participants with a positively biased forecast set expected to perform better than those with negatively biased forecast sets, due to the protection and damage costs, we have decided to not mention it as it is not true. Indeed, Figure 3 shows the final purse that participants would have, had they followed the median of their forecasts for all the cases of round 1. From this figure, it is evident that blue river participants with positively biased forecasts are expected to end the first round with less tokens in their purse than blue river participants with negatively biased forecasts. This is because of the number of misses and false alarms for both sets of participants, presented in Table 2.
- The terms "negatively biased forecasts", "unbiased forecasts" and "positively biased forecasts" were used instead of the terms "forecast set type 1-3" in the text, tables and figures. This makes the paper clearer for the reader and was also mentioned by referee 2.
- The participants' ratings of their forecast set quality and decision-maker performance were mentioned using the terms 'very bad' to 'very good', instead of rating 1 to 5 in the text, tables and figures. This makes the paper clearer for the reader.
- The language and grammar has been thoroughly revised and small changes made to improve the flow of the paper (including section titles, such as the title of sections 2, 2.1 and 3.6).

[revised manuscript text omitted]